# RALYL increases hepatocellular carcinoma stemness by sustaining the mRNA stability of TGF-β2

Xia Wang[1,2,3], Jin Wang[4], Yu-Man Tsui[2,3], Chaoran Shi[1,2], Ying Wang[1,2,5], Xin Zhang[2,3], Qian Yan[1,2], Miao Chen[1,2], Chen Jiang[1,6,7], Yun-Fei Yuan[6], Chun-Ming Wong[2,3], Ming Liu[1,2,8], Zeng-yu Feng[9], Honglin Chen[10], Irene Oi Lin Ng[2,3], Lingxi Jiang[2,9 ✉] & Xin-Yuan Guan[1,2 ✉]

Growing evidences suggest that cancer stem cells exhibit many molecular characteristics and phenotypes similar to their ancestral progenitor cells. In the present study, human embryonic stem cells are induced to differentiate into hepatocytes along hepatic lineages to mimic liver development in vitro. A liver progenitor specific gene, RALY RNA binding protein like (*RALYL*), is identified. *RALYL* expression is associated with poor prognosis, poor differentiation, and metastasis in clinical HCC patients. Functional studies reveal that *RALYL* could promote HCC tumorigenicity, self-renewal, chemoresistance, and metastasis. Moreover, molecular mechanism studies show that *RALYL* could upregulate TGF-β2 mRNA stability by decreasing N6-methyladenosine (m⁶A) modification. TGF-β signaling and the subsequent PI3K/AKT and STAT3 pathways, upregulated by RALYL, contribute to the enhancement of HCC stemness. Collectively, *RALYL* is a liver progenitor specific gene and regulates HCC stemness by sustaining TGF-β2 mRNA stability. These findings may inspire precise therapeutic strategies for HCC.

---

[1] Department of Clinical Oncology, The University of Hong Kong, Hong Kong, China. [2] State key Laboratory of Liver Research, The University of Hong Kong, Hong Kong, China. [3] Department of Pathology, The University of Hong Kong, Hong Kong, China. [4] School of Biomedical Sciences, Li Ka Shing Faculty of Medicine, The University of Hong Kong, Hong Kong, China. [5] Department of Radiation Oncology, Sun Yat-Sen University Cancer Center, Guangzhou, China. [6] State Key Laboratory of Oncology in Southern China, Sun Yat-Sen University Cancer Center, Guangzhou, China. [7] Department of Pathology, Sun Yat-Sen University Cancer Center, Guangzhou, China. [8] Affiliated Cancer Hospital and Institute of Guangzhou Medical University, Guangzhou Municipal and Guangdong Provincial Key Laboratory of Protein Modification and Degradation, School of Basic Medical Sciences, Guangzhou Medical University, Guangzhou, China. [9] Department of General Surgery, Ruijin Hospital, Shanghai JiaoTong University School of Medicine, Shanghai, China. [10] Department of Microbiology, Li Ka Shing Faculty of Medicine, The University of Hong Kong, Hong Kong, China. ✉email: jlx12120@rjh.com.cn; xyguan@hku.hk

In the last decade, an emerging body of evidence has supported the notion that tumors are hierarchically organized, and may contain a small subpopulation of cells with different biologic features and capabilities to drive tumor initiation and metastasis; these cells are called cancer stem cells (CSCs). Evidence shows that CSCs are responsible for the initiation and development of tumors and are also endowed with tissue progenitor cell features, such as the capabilities to maintain self-renewal and differentiation[1–3]. Signaling pathways regulating normal stem cells development are showed to be associated with cancer development and oncogenesis, such as bcl-2, c-Myc, transforming growth factor beta (TGF-β), Notch, Hedgehog, and Wnt signaling[4–7]. The similarities between CSCs and tissue progenitor cells suggest that understanding the molecular mechanism of normal stem cells could provide insights into CSCs, thereby facilitate the identification of CSC targets for cancer treatment.

Hepatocellular carcinoma (HCC) is one of the most lethal and prevalent cancers worldwide[8,9]. The high mortality rate of HCC is primarily because of its high tumor recurrence and metastasis rates even after eradicating its primary lesion[10]. The concept of CSCs maybe responsible for clinical observations such as tumor recurrence, metastasis, tumor dormancy, and chemoresistance capability[11]. Clinically, HCCs expressing markers of liver progenitor cells such as alpha-fetoprotein (AFP), cytokeratin 7 (CK7), cytokeratin 19 (CK19), and SOX9 that are usually turned out poor outcome and high recurrence rate[12–14]. Further molecular study showed that liver CSCs and liver progenitor cells share similar gene expression patterns such as *AFP*, *CK19*, *EpCAM*, *CD133,* and *CD90*[13]. Thus, investigation of molecular events of liver progenitor cells might help understanding the tumorigenesis in human liver.

Some studies have attempted to induce hepatocytes from several sources, such as embryonic, fetal, and somatic stem cells, by recapitulating the pathways controlling liver development[15,16]. Here, we reported that definitive endoderm (DE), liver progenitor (LP) cells, and premature hepatocytes (PH) could be induced from human embryonic stem (ES) cells by treating cell cultures with certain concentrations of factors[17]. By deep RNA sequencing, we characterized molecular signatures of liver progenitor cells and premature hepatocytes, which may be important to HCC development. Network analysis have revealed several novel specific biomarkers and potential oncogenic drivers, of which, a liver progenitor cell-specific gene, RALY RNA binding protein-like (*RALYL*) was investigated in this study. RALYL is a member of the heterogeneous nucleus ribonucleoprotein (hnRNP) family, which is constituted by RNA-binding proteins involved in transcriptional and post-transcriptional regulation[18–23]. RALYL in particular shows high homology to RALY (hnRNP associated with lethal yellow) and hnRNPC (heterogeneous nuclear ribonucleoproteins C) in the RNA recognition motif[24]. RALY and hnRNPC are reported to regulate the stability of specific transcripts[23–27]. However, the molecular function of RALYL remains unclear. In the present study, we found that RALYL could increase the stemness of HCC by improving the TGF-β2 mRNA stability via decreasing N6-methyladenosine (m[6]A) modification, which is one of the most common mRNA modification regulating multiple aspects of mRNA biology such as mRNA decay and translation[28,29]. The roles of *RALYL* in liver CSCs may provide potential oncogenic driver, ideal for HCC therapeutic targets.

## Results

### Establishment of an in vitro hepatocyte differentiation model.
An in vitro hepatocyte differentiation model was established from ES cells into DE, LP cells, and PH cells (Fig. 1a). Transcriptome sequencing was performed to identify differential expression genes among cells of the four developmental stages together with two normal liver specimens and two HCC clinical samples (NL and HCC). The heatmap of the expression profiles for the selected marker genes for ES, DE, LP, and hepatocytes showed the reliability of this differentiation model (Fig. 1b). As expected, the specific markers were indeed highly expressed in their corresponding stages. qRT-PCR also demonstrated that these specific markers, including ES markers (*OCT4* and *SOX2*), DE markers (*SOX17* and *FOXA2*), LP markers (*CK19* and *AFP*), and hepatocyte markers (*ALB* and *CYP3A4*), were highly expressed in their corresponding stages (Supplementary Fig. 1a). Expression pattern of HCC stemness related markers *CD133*, *c-Myc*, and *EpCAM* were also tested by qRT-PCR in our in vitro liver development model. The results showed that these genes were highly expressed in LP and PH stages (Supplementary Fig. 1a). To identify genes involving stemness regulation, a group of genes encoding nuclear protein that highly expressed in LP stage (Supplementary Fig. 1b) were transiently transfected into MIHA and PLC-8024 cells. Among those genes, *RALYL* was selected for further study because it upregulated stemness-related markers (e.g., *CD133*, *AFP*, *NANOG*, *SOX2*, *EpCAM*, and *c-Myc*) (Supplementary Fig. 1c) and possessed strong ability of spheroid formation (see below). Overexpression of mouse *RALYL* (*mRALYL*) could also upregulate stemness-related genes expression in mouse hepatocyte (Supplementary Fig. 1c). As CD133 is a well-known CSC marker in HCC, CD133[+], and CD133[−] cells were separated from Huh7 and Hep3B cells using fluorescence-activated cell sorting (FACS) separately. The expression level of *RALYL* was significantly higher in CD133[+] Huh7 and Hep3B cells, compared with CD133[−] cells (Fig. 1c). Furthermore, the subpopulation of CD133[+] cells was significantly increased in *RALYL* transfected cells than that in the control cells (*P* < 0.01, Fig. 1d). To further assess whether RALYL and CD133 were co-expressed in cells, immunofluorescence (IF) staining was performed in PLC-8024 and MIHA cells transfected by *RALYL* or control plasmid. The results revealed that CD133 was upregulated in most *RALYL*-overexpressing cells (Fig. 1e).

**Clinical significance of *RALYL* in HCC**. Expression of *RALYL* in immortalized liver cell lines and HCC cell lines were examined using RT-PCR and western blotting. The results showed that *RALYL* was expressed in high level in Huh7, Hep3B, and H2M cells (Supplementary Fig. 1d). Immunofluorescence results showed that the endogenous *RALYL* protein was mainly localized to the nucleus in Huh7, Hep3B, PLC-8024, and LO2 cells (Supplementary Fig. 1e). Expression of *RALYL* was also tested in 117 pairs of HCCs using qRT-PCR. Based on *RALYL* expression level, HCC patients were divided into two groups: *RALYL* expression group (31/117) and *RALYL* absence group (86/117). Clinical association study found that *RALYL* expression was significantly correlated with poorer differentiation state (Pearson $\chi^2$ test, *P* = 0.044), cirrhosis (Pearson $\chi^2$ test, *P* = 0.047), vascular invasion (Pearson $\chi^2$ test, *P* = 0.020), metastasis (Pearson $\chi^2$ test, *P* = 0.032), and the expression of CD133 (Pearson $\chi^2$ test, *P* = 0.003) (Table 1). Kaplan–Meier survival analysis showed that the expression of *RALYL* was significantly associated with worse overall survival (OS) and disease-free survival (DFS) rates (Fig. 1f). Similar results were also observed in TCGA database (Fig. 1f). Cox proportional hazard regression analysis showed that *RALYL* expression was an independent prognostic factor for overall survival of HCC patients (*P* = 0.047; Table 2).

**RALYL has strong tumorigenic ability**. To investigate the role of *RALYL* in tumorigenicity, *RALYL* was cloned into a lentiviral vector and stably transfected into PLC-8024, MIHA, and LO2

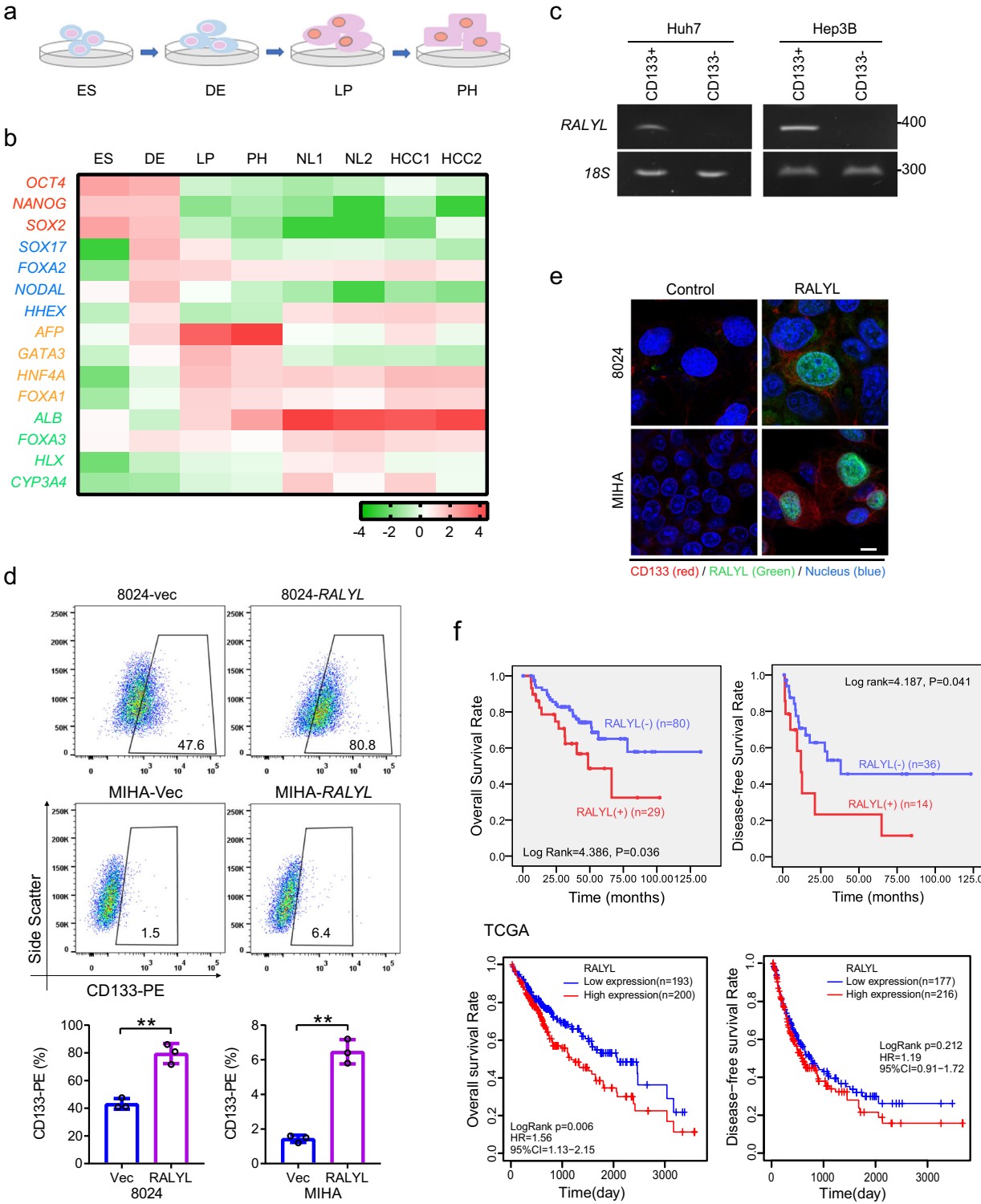

**Fig. 1 *RALYL* selection and clinical significance. a** An in vitro hepatocyte differentiation model, which induced human embryonic stem (ES) cells into definitive endoderm (DE), liver progenitor cells (LP) and premature hepatocytes (PH) step by step, was established. Two HCC samples and two normal liver tissues (NL) were also used to perform the transcriptome sequencing. **b** Heatmap of expression profiles for the specific markers for ES (*Oct4*, *Nanog*, and *Sox2*), DE (*Sox17*, *FOXA2*, *Nodal*, and *Hhex*), liver progenitor cells (*AFP*, *Gata3*, *HNF4α*, and *FOXA1*), and hepatocyte (*Albumin*, *FOXA3*, *Hlx*, and *Cyp3a4*) shows the reliability of the hepatic differentiation model. The red represents a higher expression level, and the green represents a lower expression level. **c** RT-PCR analysis showed higher *RALYL* expression in CD133[+] cells than in CD133[−] cells, which was sorted by FACS from Huh7 or Hep3B cells. **d** The proportions of CD133[+] cells in 8024-Vec/*RALYL* and MIHA-Vec/*RALYL* were assessed by flow cytometry and illustrated in bar chart. The values indicate the mean ± standard deviation (SD) of three independent experiments (**$P < 0.01$, Student *t*-test). **e** Representative images of double immunofluorescence staining of RALYL (green) and CD133 (red) in 8024-Vec/*RALYL* and MIHA-Vec/*RALYL*. DAPI (blue) was used for nuclei counterstaining. Scale bar = 10 μm. **f** Kaplan–Meier overall survival curve and disease-free survival curve of two HCC groups in our in-house cohort or The Cancer Genome Atlas (TCGA) cohort: *RALYL*(+) (red), patients with *RALYL* expression; *RALYL*(−) (blue), patients without *RALYL* detection.

**Table 1 Association between *RALYL* expression and clinicopathologic features in 117 HCC cases.**

| Features | Total | RALYL expression[a] | | P-value |
|---|---|---|---|---|
| | | **Absent** | **Present** | |
| *Sex* | | | | 0.985 |
| Male | 98 | 72 | 26 | |
| Female | 19 | 14 | 5 | |
| *Age (years)* | | | | 0.812 |
| ≤60 | 96 | 71 | 25 | |
| >60 | 21 | 15 | 6 | |
| *Serum AFP (ng/mL)* | | | | 0.075 |
| ≤400 | 65 | 52 | 13 | |
| >400 | 52 | 34 | 18 | |
| *Serum HBsAg* | | | | 0.643 |
| Negative | 16 | 11 | 5 | |
| Positive | 101 | 75 | 26 | |
| *Cirrhosis* | | | | **0.047** |
| Absent | 36 | 22 | 14 | |
| Present | 80 | 63 | 17 | |
| *Differentiation* | | | | **0.044** |
| Well/moderate | 67 | 54 | 13 | |
| Poor | 50 | 32 | 18 | |
| *Tumor size* | | | | 0.205 |
| ≤5 | 49 | 39 | 10 | |
| >5 | 68 | 47 | 21 | |
| *TNM stage (AJCC)* | | | | 0.559 |
| I | 84 | 63 | 21 | |
| II/III | 33 | 23 | 10 | |
| *Vascular invasion* | | | | **0.020** |
| Absent | 102 | 78 | 24 | |
| Present | 13 | 6 | 7 | |
| *Metastasis* | | | | **0.032** |
| Absent | 57 | 47 | 10 | |
| Present | 60 | 39 | 21 | |
| *CD133* | | | | **0.003** |
| Low | 78 | 64 | 14 | |
| High | 39 | 22 | 17 | |
| *TGF-β2* | | | | **0.029** |
| Absent | 72 | 58 | 14 | |
| Present | 45 | 28 | 17 | |

Pearson $\chi^2$ test.
Statistical significance ($P < 0.05$) is shown in bold.
#Partial data are not available, and the statistic was based on available.
[a]*RALYL* absent: Samples without *RALYL* detection by qRT-PCR; *RALYL* present: Samples detected with *RALYL* expression by qRT-PCR.

cells. The ectopic expression of *RALYL* was confirmed in both mRNA and protein levels (Fig. 2a and Supplementary Fig. 2a). XTT proliferation assays showed that the overexpression of *RALYL* promoted tumor cell proliferation (Fig. 2b and Supplementary Fig. 2b). In contrast to control cells, *RALYL*-overexpressing cells showed higher foci formation frequencies (Fig. 2c and Supplementary Fig. 2c) and colony formation capacity in soft agar (Fig. 2d and Supplementary Fig. 2c), indicating that *RALYL* could significantly enhance tumor cell growth in both anchorage-dependent and anchorage-independent manners. In addition, *RALYL* expression was silenced in Huh7, H2M, and Hep3B cells with two short hairpin RNAs (shRNA) (Fig. 2a and Supplementary Fig. 2a). As expected, *RALYL* silencing significantly suppressed cell proliferation (Fig. 2b and Supplementary Fig. 2b), foci formation efficiencies (Fig. 2c and Supplementary Fig. 2c), and colony formation in soft agar (Fig. 2d and Supplementary Fig. 2c).

The mouse xenograft model, after subcutaneous injection, showed that tumors induced by *RALYL*-overexpressing cells were much larger than those induced by control cells (Fig. 2e and Supplementary Fig. 2d). Furthermore, the frequency of tumor formation was decreased in cells with *RALYL* silencing (1/6 in Huh7 and 3/5 in Hep3B), compared with scrambled shRNA-transfected cells (6/6 in Huh7 and 5/5 in Hep3B) (Fig. 2e). The tumor volume was also significantly smaller in tumors induced by sh*RALYL*-transfected cells than in those induced by control cells (Fig. 2e).

***RALYL* promotes cell motility and metastasis by inducing EMT.** The effect of *RALYL* on tumor invasion and metastasis was characterized by cell migration, invasion, and in vivo liver metastasis assays. Cell migration and invasion assays revealed that overexpression of *RALYL* significantly ($P < 0.05$) enhanced HCC cell motility (Fig. 3a and Supplementary Fig. 3a). As expected, *RALYL* silencing significantly ($P < 0.05$) prevented HCC cell motility (Fig. 3b and Supplementary Fig. 3b). The in vivo liver metastatic model, 10 weeks after intrasplenic injection, showed that all five mice injected with *RALYL*-transfected PLC-8024 cells had metastatic nodules on their liver surfaces, whereas only a few

**Table 2 Cox proportional hazard regression analyses for 5-year survival.**

| Clinicopathological features | Multivariate analysis | | | Un-variate analysis | | |
|---|---|---|---|---|---|---|
| | HR | 95% CI | P value | HR | 95% CI | P value |
| *Age* | | | | | | |
| <60 years vs. >60 years | 0.487 | 0.316–1.728 | 0.485 | 0.079 | 0.444–2.960 | 0.778 |
| *Gender* | | | | | | |
| Male vs. female | 0.239 | 0.220–2.483 | 0.625 | 0.264 | 0.268–2.159 | 0.607 |
| *TNM stage* | | | | | | |
| I vs. II and III | 0.434 | 0.281–1.879 | 0.510 | 11.751 | 0.158–0.605 | **0.001** |
| *Tumor size* | | | | | | |
| ≤5 cm vs. >5 cm | 0.985 | 0.718–2.753 | 0.321 | 0.659 | 0.379–1.495 | 0.417 |
| *Differentiation* | | | | | | |
| Poor vs. well | 4.579 | 0.192–0.930 | **0.032** | 7.244 | 0.164–0.752 | **0.007** |
| *Cirrhosis* | | | | | | |
| Absent vs. present | 0.322 | 0.282–2.009 | 0.570 | 2.579 | 0.852–4.993 | 0.108 |
| *Vascular invasion* | | | | | | |
| Absent vs. present | 3.283 | 0.115–1.089 | 0.070 | 21.503 | 0.291–13.916 | **<0.001** |
| *RALYL expression* | | | | | | |
| Absent vs. present | 3.936 | 0.245–0.992 | **0.047** | 4.204 | 0.344–0.969 | **0.040** |
| *CD133 upregulation* | | | | | | |
| No vs. yes | 4.670 | 0.218–0.928 | **0.031** | 5.057 | 0.236–0.906 | **0.025** |
| *TGF-β2 upregulation* | | | | | | |
| No vs. yes | 0.369 | 0.347–1.746 | 0.543 | 9.186 | 0.183–0.695 | **0.002** |

Cox regression analysis. Statistical significance ($P < 0.05$) is shown in bold. CI, confidence interval; HR, hazard ratio.
*CI* confidence interval, *HR* hazard ratio.

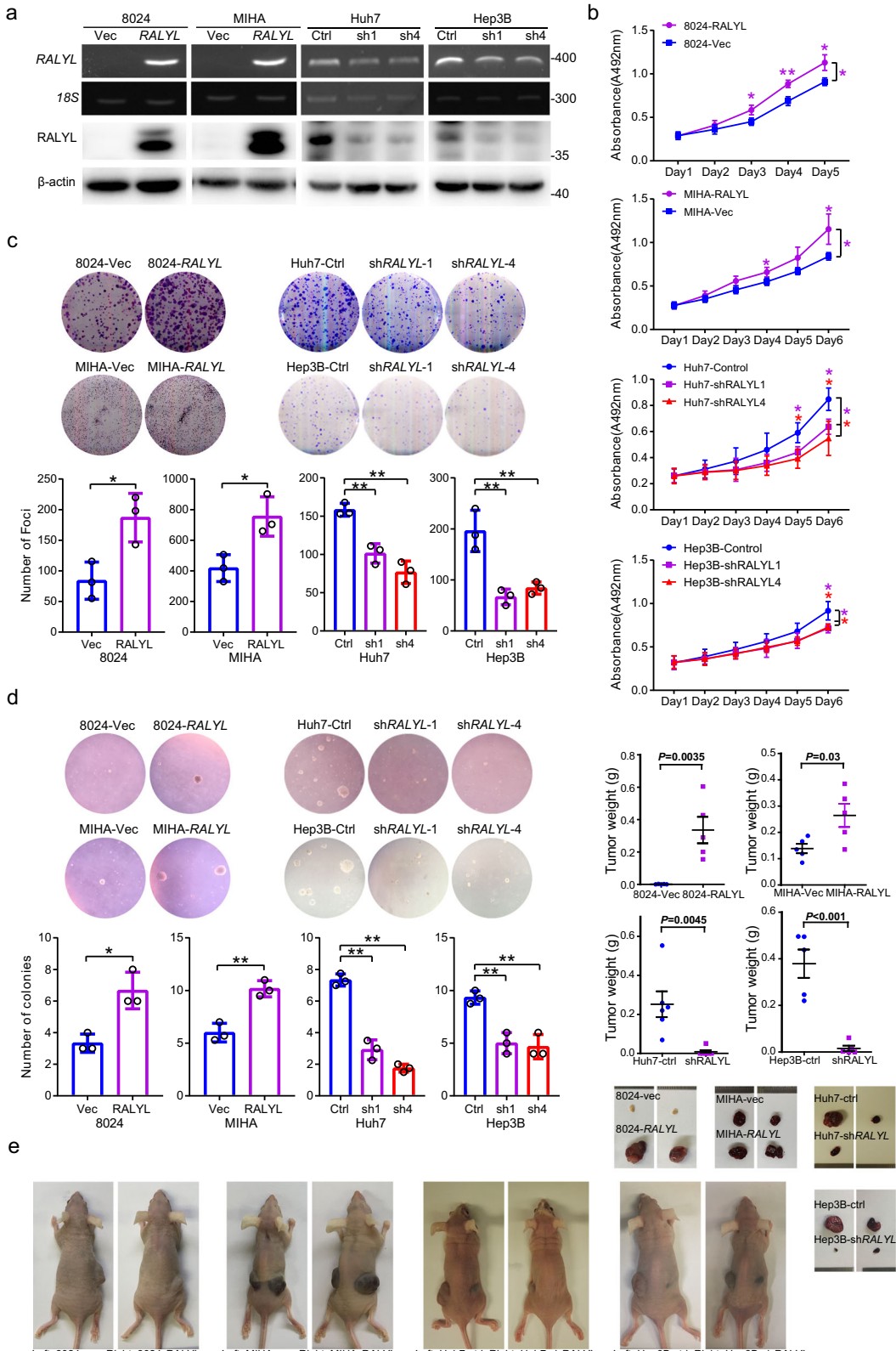

metastatic nodules were observed in 2/4 of mice in the control group (Fig. 3c). In addition, no metastatic nodules were observed in sh*RALYL*-transfected cells, whereas multiple metastatic nodules were formed on the liver surfaces in the control group (Fig. 3c). H&E staining was used to further confirm the liver metastasis lesions (Fig. 3c). Taken together, these findings strongly suggested that *RALYL* could promote HCC metastasis.

During the epithelial–mesenchymal transition (EMT) process, epithelial cells acquired migratory and invasive mesenchymal properties, which contributes to tumor metastasis. Western blotting revealed that *RALYL* could downregulate the expression of epithelial marker (E-cadherin) and upregulate the mesenchymal markers (N-cadherin and Fibronectin) (Fig. 3d). As expected, *RALYL* knockdown could upregulate epithelial markers

**Fig. 2 *RALYL* shows strong tumorigenic ability. a** RT-PCR and western blotting showed ectopic expression of *RALYL* in cells transfected with *RALYL* or control plasmid, and *RALYL* silencing in cells treated with scrambled shRNA (Control: Ctrl) or shRNA against *RALYL* (sh*RALYL*-1: sh1; sh*RALYL*-4: sh4). 18S and β-actin were used as loading controls. **b** XTT assay was used to determine the cell proliferation rates. The values indicate the mean ± SD of three independent experiments. *P* value was shown as *($P < 0.05$) or **($P < 0.01$) at some time points and highlighted in the color same with the line of corresponding experiment group (* in purple: 8024-*RALYL* vs. 8024-Vec, MIHA-*RALYL* vs. MIHA-Vec, Huh7-sh*RALYL*1 vs. Huh7-Control, Hep3B-sh*RALYL*1 vs. Hep3B-Control; * in red: Huh7-sh*RALYL*4 *vs* Huh7-Control, Hep3B-sh*RALYL*4 vs. Hep3B-Control, independent Student's *t*-test). Representative images of foci formation (**c**) and colony formation in soft agar (**d**) induced by 8024-Vec/*RALYL*, MIHA-Vec/*RALYL*, Huh7-Control/sh*RALYL* and Hep3B-Control/ sh*RALYL*. The numbers of foci and colonies are illustrated in bar chart. The values indicate the mean ± SD of three independent experiments (*$P < 0.05$; **$P < 0.01$, two-sided Student's *t*-test). **e** Representative images of mice with tumors induced by the indicated cells. Weights of tumors are expressed as mean ± SD of five mice (independent Student's *t*-test).

and downregulate mesenchymal markers. EMT-related transcription factors, such as SNAIL and SLUG, were also upregulated in *RALYL*-overexpressing cells and downregulated in *RALYL*-silencing cells, indicating that *RALYL* could promote EMT (Fig. 3d).

**RALYL enhances the stemness characteristics of HCC.** As *RALYL* is specifically expressed in liver progenitor cells and premature hepatocytes stages, we hypothesized that *RALYL* might regulate HCC stemness. Next, the effects of *RALYL* on cancer stemness were investigated by both in vitro and in vivo assays. First, western blotting results indicated that *RALYL* overexpression could upregulate stemness-related markers AFP, CD133, NANOG, SOX2, and c-Myc (Fig. 4a and Supplementary Fig. 4a), whereas *RALYL* silencing decreased the expression of these genes (Fig. 4a and Supplementary Fig. 4a). Second, the spheroid formation assay showed that the upregulation of *RALYL* significantly ($P < 0.05$, Student's *t*-test) enhanced primary and secondary spheroid formation frequencies, compared with control cells (Fig. 4b and Supplementary Fig. 4b), indicating that *RALYL*-expressing cells had high self-renewal ability. Expectedly, *RALYL*-silencing cells formed lesser and smaller spheroids than the control cells (Fig. 4b and Supplementary Fig. 4b). Moreover, the *mRALYL* was overexpressed in mouse liver organoid. Results revealed that more and larger mouse liver organoids were observed in *mRALYL* group compared to control group (Supplementary Fig. 4c). Third, the in vivo tumor formation experiment revealed that as few as 50,000 *RALYL*-overexpressing cells were sufficient to generate tumors in nude mice, whereas no tumor was formed when mice were injected with the same amount of control cells (Fig. 4c). To investigate the effect of *RALYL* on cell differentiation, all-trans retinoic acid (atRA) was used to treat cells to induce differentiation. qRT-PCR results found that the fold changes of expressions of mature hepatocyte markers (*CK8*, *CK18*, and *albumin*) were significantly higher in 8024-*RALYL* and MIHA-*RALYL* cell after atRA treatment, compared with 8024-Vec and MIHA-Vec. Whereas, the fold change of expressions of stemness-associated genes (*AFP*, *CD133*, and *NANOG*) and *RALYL* were significantly lower in 8024-*RALYL* and MIHA-*RALYL* cells compared with controls (Fig. 4d), suggesting that *RALYL*-expressing cells possessed higher differentiation potential.

**RALYL enhances the chemoresistance of HCC cells.** Resistance to chemotherapeutic agents is a typical stemness-related property. After treating with two commonly used chemotherapeutic reagents, cisplatin (CDDP) and 5-fluorouracil (5-Fu), at different concentrations, cell viability of *RALYL*-overexpressing cells was significantly higher than that of controls (Fig. 4e and Supplementary Fig. 5a). Flow cytometry results also revealed that the percentage of apoptotic cells was significantly lower in *RALYL*-transfected cells than in controls (Supplementary Fig. 5b).

Consistently, TUNEL assay results confirmed that *RALYL*-overexpressing PLC-8024 cells had significantly lower apoptotic index than control cells (Supplementary Fig. 5c). As expected, *RALYL* silencing had opposite effects on the chemoresistance of HCC (Fig. 4e and Supplementary Fig. 5a–c). However, *RALYL* did not enhance resistance to Sorafenib (Supplementary Fig. 5a).

To further confirm that *RALYL* can enhance the chemoresistance of HCC in vivo, nude mice with xenograft tumors, induced by 8024-Vec or 8024-*RALYL* cells, were treated with CDDP (3 mg/kg body weight) or 5-Fu (50 mg/kg body weight) when the tumors at each side reached a similar size about 5 mm in diameter (about 8–10 days). CDDP and 5-Fu were adopted by intraperitoneal injection every four days. The xenograft tumors in the 8024-*RALYL* group grew faster and larger than those in the control group (Fig. 4f). Interestingly, we found that the *RALYL*-expressing cells were significantly enriched in PLC-8024-induced tumor tissues treated with chemotherapeutic reagents (Fig. 4f), indicating that they are more chemoresistant.

**RALYL enhances stemness of HCC through TGF-β2 signaling.** To characterize the underlying molecular mechanism of *RALYL* in stemness regulation, transcriptome sequencing was performed to compare expressing profiles between 8024-*RALYL* and 8024-Vec cells. Approximately 450 and 800 genes were upregulated and downregulated in 8024-*RALYL* cells, respectively. Gene ontology (GO) analysis revealed that most of the upregulated genes were enriched in the transcription, RNA biosynthetic, and RNA metabolic processes (Supplementary Fig. 6a). KEGG enrichment analysis showed that the upregulation of *RALYL* could regulate TGF-β signaling and increase the expression of TGF-β2 (Supplementary Fig. 6b). qRT-PCR results confirmed that *TGF-β2* was upregulated in *RALYL*-overexpressing cells (8024-RALYL, MIHA-RALYL, and LO2-RALYL) and downregulated in *RALYL*-silencing cells (Huh7-shRALYL, H2M-shRALYL, and Hep3B-shRALYL) (Supplementary Fig. 6c). Western blotting further showed that TGF-β2 secretion in cell culture supernatants was increased in *RALYL*-overexpressing cells, compared to controls (Fig. 5a). Consistently, *RALYL* overexpression promoted the expression of TGF-β2 in HCC cells, and *RALYL* silencing had opposite results (Fig. 5b and Supplementary Fig. 6d). Previous studies indicated that TGF-β could regulate several stemness-related pathways, such as PI3K/AKT and STAT3 pathways[30,31]. As expected, *RALYL* overexpression increased the levels of AKT, STAT3 and phosphorylated PI3K, AKT, and STAT3 (Fig. 5b and Supplementary Fig. 6d). Consequently, the targets of STAT3, including NANOG, c-Jun, c-Myc, and BCL-XL, which played important roles in stemness of cancer, were also upregulated in *RALYL*-overexpressing cells. *RALYL* knockdown experiments revealed consistent results (Fig. 5b and Supplementary Fig. 6d).

To further validate the hypothesis that HCC stemness could be enhanced through the increased production of TGF-β2 regulated by *RALYL*, PLC-8024, or MIHA cells were treated with a

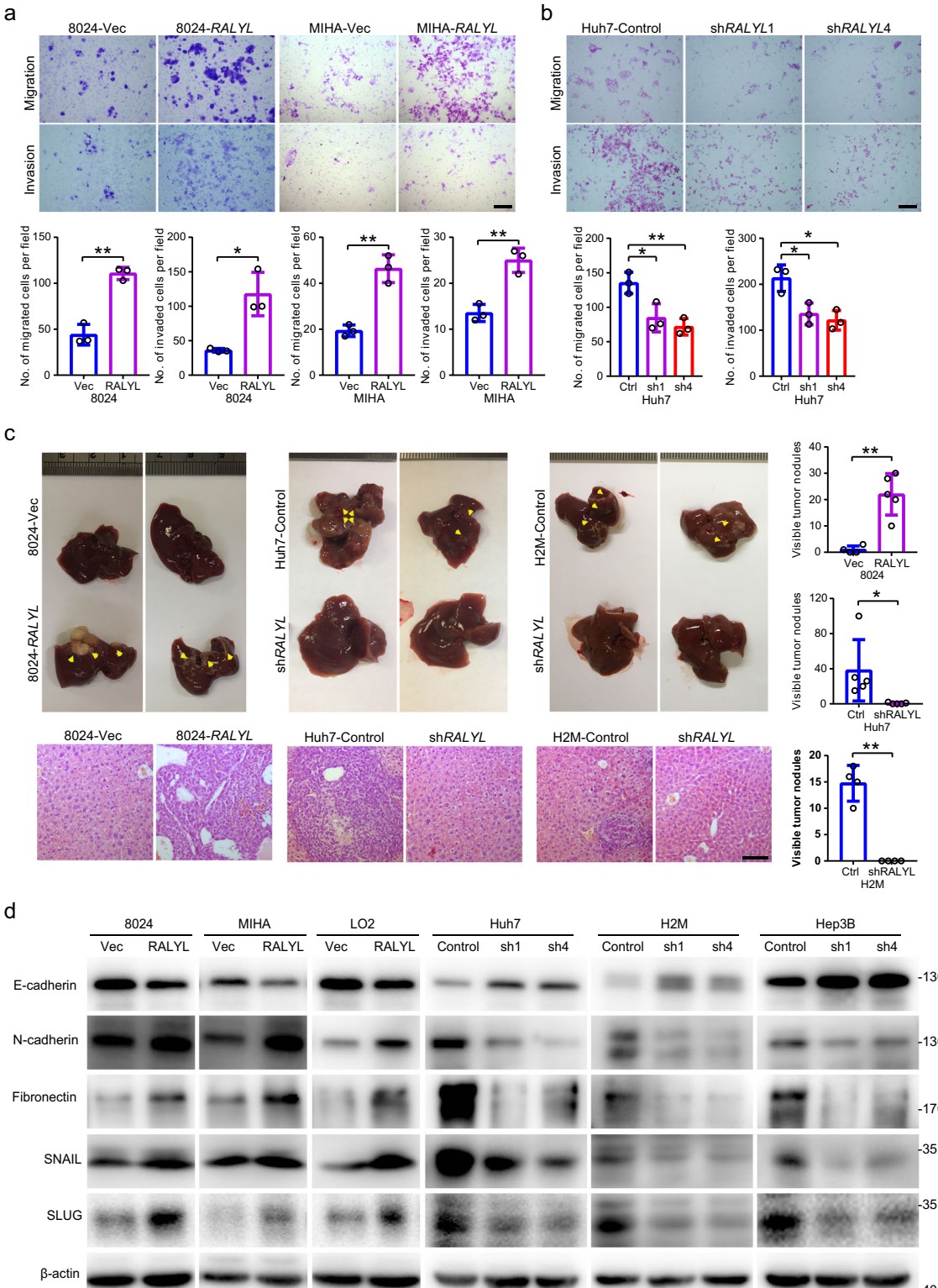

**Fig. 3 RALYL promotes HCC migration, invasion and metastatic ability.** Cell motilities of 8024-Vec/*RALYL*, MIHA-Vec/*RALYL* (**a**) and Huh7-Control/ sh*RALYL*s (**b**) were assessed by transwell migration and matrigel invasion assays. The number of migrated or invaded cells is shown in the bar chart. The values represent the mean ± SD of three independent experiments (*P < 0.05, **P < 0.01, two-sided Student's *t*-test). Scale bar = 200 μm. **c** Representative images of livers derived from nude mice after intrasplenic injection of indicated cells (up) and representative H&E staining of the corresponding liver sections (down). Scale bar = 200 μm. The number of metastatic nodules on the liver surface (indicated by yellow arrows) is summarized in the bar charts (right) and represent mean ± SD of four mice of PLC-8024-Vec, five mice of PLC-8024-RALYL, five mice of Huh7-Control or shRALYL, and four mice of H2M-Control or shRALYL (*P < 0.05, **P < 0.01, independent Student's *t* test). **d** The protein expression level of E-cadherin, N-cadherin, fibronectin, SNAIL, and SLUG of cells transfected with Vec and *RALYL*, or Control and sh*RALYL*s.

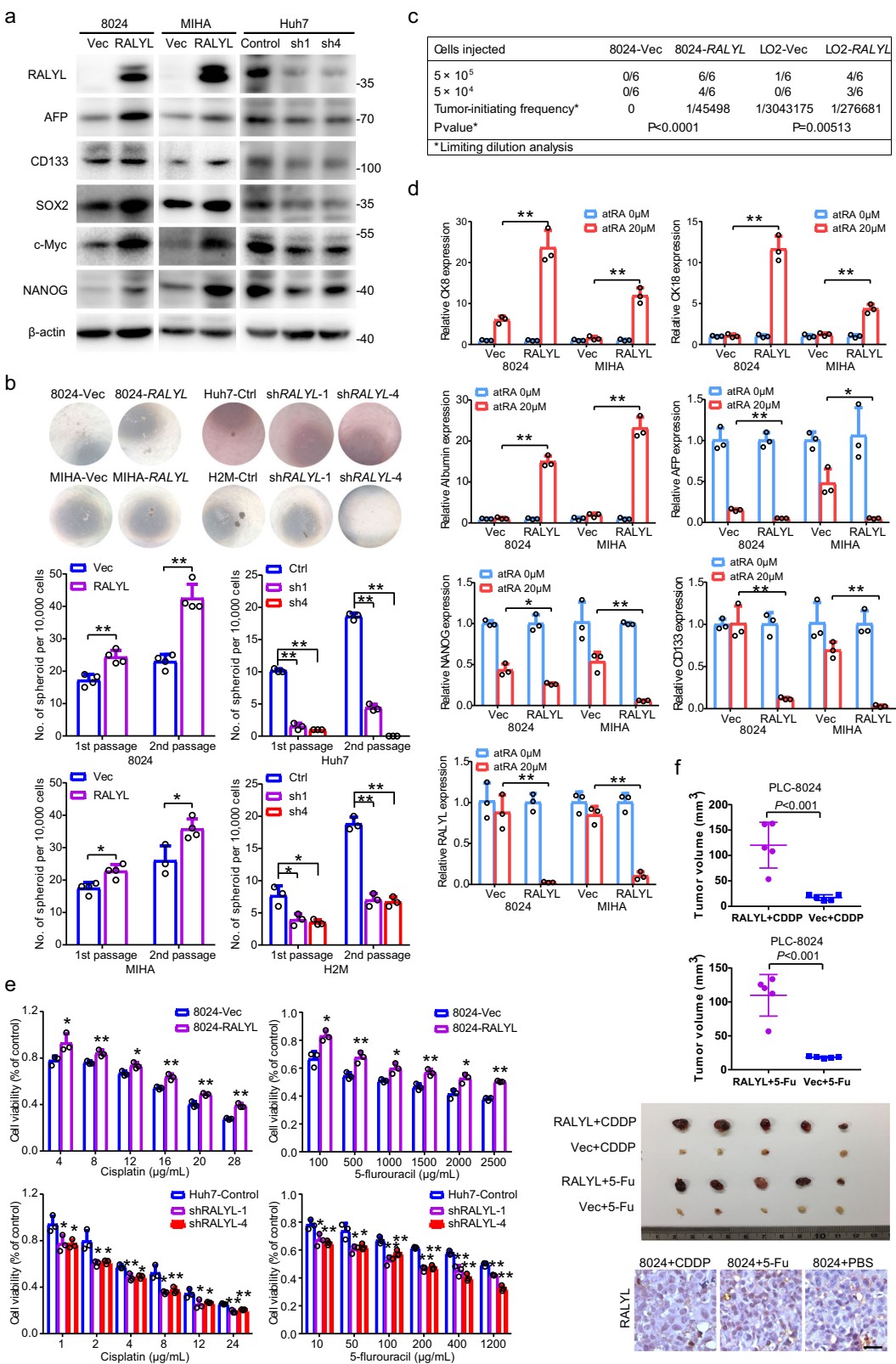

conditioned medium (CM) from *RALYL*-overexpressing or control cells, respectively. The recombinant human TGF-β2 protein was also used to treat PLC-8024 and Huh7-sh*RALYL* cells. As expected, cells treated with CM from *RALYL*-expressing cells and TGF-β2 proteins showed increased levels of phosphorylation of AKT and STAT3 and the downstream targets c-Myc and c-Jun (Fig. 5c, d), whereas, the TGF-β2 neutralization

antibody (1 μg/mL) could abolish the activation of AKT and STAT3 induced by *RALYL* in PLC-8024 (Fig. 5d). To confirm the major role of STAT3 in *RALYL*-mediated TGF-β2 signaling, a STAT3 specific inhibitor, NSC74859, was used to treat HCC cells. The results revealed that cells with higher expression levels of *RALYL* were more sensitive to NSC74859 (Fig. 5e). Mouse xenograft models also showed consistent results (Fig. 5f). Taken

**Fig. 4 RALYL enhances stemness properties of HCC cells. a** Western blotting was performed to determine the expression of stemness markers in *RALYL*-overexpressing cells and *RALYL*-silencing cells. **b** Spheroid formation assay was used to evaluate the self-renewal ability of 8024-Vec/*RALYL*, MIHA-Vec/*RALYL*, Huh7-Control/sh*RALYLs* and H2M-Control/sh*RALYLs*. The numbers of primary and secondary spheroids are calculated in the bar chart below. The values represent the mean ± SD of three independent experiments (*$P < 0.05$, **$P < 0.01$, independent Student's *t*-test). **c** Limiting dilution assay shows tumor-initiating frequency of 8024-Vec/*RALYL* or LO2-Vec/*RALYL* cells in nude mice. **d** qRT-PCR was used to compare fold change of expressions of stemness-related and liver differentiation-related markers in HCC cells after treating with atRA for 5 days. The values represent the mean ± SD of three independent experiments (*$P < 0.05$, **$P < 0.01$, independent Student's *t*-test). **e** XTT assay reveals stronger chemoresistance ability of cells with higher *RALYL* expression compared with that of lower *RALYL* expression. Cells were treated with the indicated concentration of CDDP and 5-Fu for 48 h. The values represent the mean ± SD of three independent experiments (*$P < 0.05$, **$P < 0.01$, two-sided Student's *t*-test). **f** Representative images of tumors induced by 8024-Vec/*RALYL* treated with indicated drugs (middle). The average tumor volume was expressed as mean ± SD of five mice. Representative images of RALYL IHC staining of PLC-8024-induced tumors treated with indicated reagents (lower right). Scale bar = 32 μm.

together, *RALYL* could regulate HCC stemness through STAT3-dependent TGF-β2 upregulation.

**RALYL binds to TGF-β2 mRNA and improves its stability in m⁶A-dependent way.** As RALYL has been reported to show high homology to RALY and hnRNPC, which were regulators of the stability of specific transcripts, we next studied whether RALYL could stabilize TGF-β2 mRNA by the treatment with actinomycin D (ActD, 5 μg/mL), an inhibitor of RNA synthesis. The results indicated that TGF-β2 mRNA was more stable in *RALYL*-transfected cells than that in control cells (Fig. 6a). Meanwhile, the stability of GAPDH mRNA, which was used as a negative control, showed no differences (Fig. 6a). The RNA immunoprecipitation (RIP) assay results revealed that TGF-β2 mRNA was significantly enriched in RALYL immune-precipitates compared with the amount of GAPDH mRNA, which served as a negative control (Fig. 6b). Moreover, TGF-β2 mRNA was almost undetected in rabbit IgG immune-precipitates (Fig. 6b), suggesting that RALYL is able to upregulate TGF-β2 by binding to its mRNA and improving its stability.

The loss of m⁶A methylation was reported to increase the stability of transcripts[32]. To investigate whether RALYL increases TGF-β2 mRNA stability through m⁶A methylation modification, an antibody recognizing m⁶A was used for RNA immunoprecipitation (meRIP) of RNA from *RALYL*-overexpressing cells or control cells. The gene-specific meRIP-qPCR showed significantly lower levels of m⁶A in TGF-β2 mRNA of the *RALYL*-overexpressing cells (Fig. 6c). HPRT1 without m⁶A modification was used as negative control[33]. As m⁶A methylation of mRNA 3′-UTR played an important role in the stability of mRNA, a luciferase reporter assay was performed to validate the interaction between RALYL and 3′-UTR of TGF-β2 mRNA. Cells were transfected with reporter plasmids containing the entire *TGF-β2* 3′-UTR. The relative luciferase activity in *RALYL*-overexpressing cells was significantly augmented compared with control cells (Fig. 6d). To further address whether other m⁶A-regulatory proteins are involved in the m⁶A modification of TGF-β2 mRNA, immunoprecipitation (IP) assay was employed. Interestingly, IP results showed that Fat mass and obesity-associated protein (FTO), a well-known m⁶A eraser, could interact with RALYL (Fig. 6e). Taken together, we concluded that RALYL could cooperate with FTO to remove m⁶A of TGF-β2 mRNA and keep its stability.

**Discussion**
HCC is one of the most fatal malignancies, mainly due to its high tumor recurrence, metastasis, and resistance to conventional chemotherapy or radiotherapy. The existence of CSCs is now well accepted, which are considered to be the fundamental cause of these clinical observations, including tumor recurrence, metastasis, and chemoresistance. Thus, understanding the molecular mechanism of CSCs will facilitate the exploitation of clinical

therapeutic strategies. Emerging evidence suggests that CSCs and tissue progenitor cells shared many common properties, which are widely applied by histopathologists to define the differentiation level of certain types of tumors[34]. HCC sharing similar gene expression pattern to liver progenitor cells usually carry a poorer prognosis[12]. In the present study, a hepatocyte differentiation model induced from ES was used to identify genes specifically expressed in liver progenitor cells, which should be important in liver development, and play crucial roles in stemness of HCC. Among them, a liver progenitor specific gene *RALYL* was investigated. Clinically, *RALYL* expression is correlated with poor prognosis ($P = 0.036$), poor differentiation ($P = 0.044$), metastasis ($P = 0.032$), CD133 expression ($P = 0.003$), and *TGF-β2* expression ($P = 0.029$) in HCC patients. The expression of *RALYL*, *CD133*, and *TGF-β2* are closely associated with poor overall prognosis in our in-house cohort ($P_{RALYL} = 0.036$, $P_{CD133} = 0.021$, and $P_{TGF-β2} = 0.002$) as well as TCGA database ($P_{RALYL} = 0.006$, $P_{CD133} = 0.004$, and $P_{TGF-β2} = 0.047$). Interestingly, *RALYL* expression was significantly higher in CD133⁺ HCC cells. Moreover, overexpression of *RALYL* could increase the proportion of CD133⁺ cells, as well as the expression of stemness-related markers, such as AFP, CD133, NANOG, and c-Myc. All these data indicate that *RALYL* may play important roles in the maintenance of HCC stemness. Functional studies revealed that *RALYL* could enhance tumorigenicity and spheroid formation of HCC cells. AtRA-treated HCC cells revealed that *RALYL*-overexpressing cells possessed higher differentiation ability. Further study showed that *RALYL*-expressing cells are more resistant to CDDP and 5-Fu both in vitro and in vivo. All these results indicate that *RALYL* could upregulate the expression of stemness-related markers; increase stemness properties of HCC, including tumorigenicity, self-renewal, chemoresistance, and maintain HCC cells in a poor differentiation state. Consistent with the clinical association between *RALYL* expression and vascular invasion and metastasis, we also found that *RALYL* could promote HCC invasion and metastasis by promoting the EMT process.

RALYL belongs to the hnRNPs family, which is composed of RNA-binding proteins that are involved in the process of transcriptional and post-transcriptional regulation[27]. Thus, we speculate that RALYL may regulate specific gene expressions by the RNA binding manner. To explore the underlying molecular mechanism of RALYL in stemness regulation, transcriptome sequencing was performed to compare different gene expressions between 8024-*RALYL* and 8024-Vec cells. In line with our speculation, GO analysis revealed that most of the upregulated genes in 8024-*RALYL* cells are involved in the transcription and RNA metabolic processes. KEGG enrichment analysis showed that RALYL could regulate TGF-β signaling and upregulate the expression of TGF-β2. RALYL is reported to be highly homologous to RALY and hnRNPC, which are reported to regulate the stability of specific transcripts[24,25,27]. In this study, we found that RALYL could enhance TGF-β2 mRNA stability by binding to its

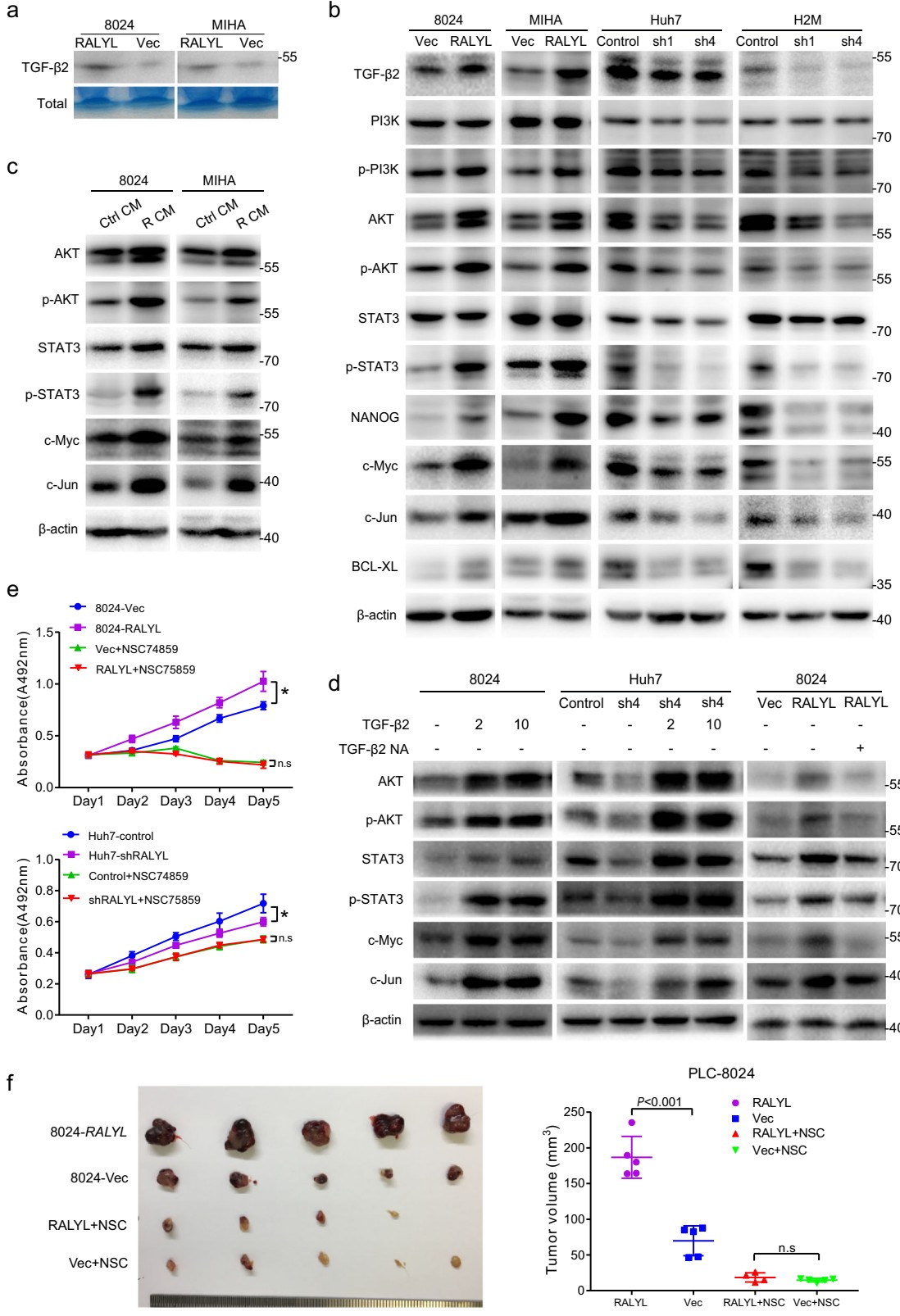

mRNA. m⁶A is a prevalent internal modification of mRNA in mammalian cells, and is reported to accelerate mRNA decay[28]. We next investigated whether RALYL increases TGF-β2 mRNA stability by modulating m⁶A modification. Interestingly, meRIP-qPCR results indicated lower level of m⁶A in TGF-β2 mRNA was observed in the *RALYL*-overexpressing cells compared with control cells. As reported, m⁶A modification usually presents in

the consensus sequence RRACH and the enrichment of m⁶A is commonly observed in 3′ untranslated regions (3′ UTRs) near stop codon, which regulates the stability of mRNA[32]. We found that multiple RRACH sites are located in 3′ UTR of TGF-β2 mRNA. Therefore, we further confirmed the interaction between RALYL and 3′-UTR of TGF-β2 mRNA by luciferase reporter assays. Moreover, we found that FTO, a well-known m⁶A eraser,

**Fig. 5 RALYL enhances HCC stemness through TGF-β2 signaling. a** The TGF-β2 secretion level in cell culture medium was confirmed by western blotting. Total proteins staining with Coomassie Brilliant Blue were used as the loading control. **b** Western blotting was performed to determine the expression of TGF-β2, PI3K, phosphorylated-PI3K (p-PI3K), AKT, phosphorylated-AKT (p-AKT), STAT3, $p^{Y705}$STAT3 (p-STAT3), NANOG, c-Myc, c-Jun, and BCL-XL in cell lysates from 8024-Vec/*RALYL*, MIHA-Vec/*RALYL*, Huh7-Control/sh*RALYL*s, and H2M-Control/sh*RALYL*s. **c** PLC-8024 and MIHA cells were treated with conditioned medium from 8024-Vec/*RALYL* or MIHA-Vec/*RALYL*, respectively (Ctrl CM: conditioned medium from Vec transfected cells; R CM: conditioned medium from *RALYL* transfected cells). Cell lysates were analyzed using western blotting to compare the expression levels of AKT, p-AKT, STAT3, p-STAT3, c-Myc, and c-Jun. **d** PLC-8024 and Huh7-sh*RALYL*4 were treated with TGF-β2 at 2 or 10 ng/mL; 8024-*RALYL* cells were treated with TGF-β2-neutralizing antibody at 1 μg/mL. The expression levels of AKT, p-AKT, STAT3, p-STAT3, c-Myc, and c-Jun were determined by western blotting. **e** The XTT assay was used to assess the effect of the STAT3 inhibitor (NSC74859) on the proliferation of 8024-Vec/*RALYL* (NSC74859: 40 μM) and Huh7-Control/sh*RALYL* cells (NSC74859: 20 μM). The values represent the mean ± SD of three independent experiments (*P < 0.05, **P < 0.01, independent Student's *t*-test). **f** Representative images of tumors induced by indicated cells in nude mice treated with NSC74859 once 3 days for 3 weeks. The average tumor volume was expressed as mean ± SD of five mice.

could also interact with RALYL and might play an important role in m6A modification of TGF-β2 mRNA. Taken together, RALYL could bind to TGF-β2 mRNA, decreased its m6A modification and increased the stability of TGF-β2 mRNA. Consequently, the expression of TGF-β2 was up-regulated and the stemness of HCC was enhanced.

TGF-β plays a crucial role in the self-renewal and maintenance of stemness in the development and differentiation processes of ES and somatic stem cells. The role of the TGF-β family in tumors is complicated, which can either suppress cell proliferation or enhance cell growth and tumor metastasis, in a cellular context depended way[35,36]. Elevated levels of plasma TGF-β in HCC patients are most prominently associated with poor prognosis in HCC[37]. We found that high expression of TGF-β2 was positively correlated with poor differentiation stage ($P = 0.009$), vascular invasion ($P = 0.014$), metastasis ($P = 0.024$) (Supplementary Table 2) as well as *RALYL* expression ($P = 0.029$, Table 1). RALYL could specifically upregulate the expression of TGF-β2, consequently activate the PI3K/AKT and STAT3 pathways, and upregulate the subsequent targets, such as NANOG, c-Myc, and c-Jun. When cells treated with CM from *RALYL*-overexpressing cells or recombinant human TGF-β2 protein, the phosphorylation of AKT and STAT3 were upregulated significantly. Whereas, TGF-β2 neutralization antibody (1 μg/mL) could prevent the activation of AKT and STAT3 induced by *RALYL* overexpression. STAT3 facilitates to maintain the pluripotential phenotype of stem cell and tumor cell proliferation and invasion[30,38]. A STAT3-specific inhibitor, NSC74859, could decrease HCC cells with high *RALYL* expression in cell proliferation and tumor growth. Altogether, *RALYL* increases HCC stemness through STAT3 dependent TGF-β2 signaling.

In summary, we explored the role of *RALYL* in HCC development and demonstrated that *RALYL* could increase the stemness of HCC through TGF-β2 signaling, which could help us to unveil the molecular mechanism of CSCs and inspire strategies targeting to CSCs. Our study also supports that the molecular mechanism of liver progenitor cells could facilitate the understanding of the molecular features of CSCs.

## Methods

**HCC clinical specimens and cell lines**. Overall, 117 pairs of primary HCC and their adjacent normal specimens were obtained from patients after hepatectomy at Sun Yat-Sen University Cancer Center (Guangzhou, China). Clinical specimens used in this study were approved by the Committee for Ethical Review of Research Involving Human Subjects at the Sun Yat-Sen University Cancer Center. Human immortalized liver cell lines, i.e., MIHA and LO2, and HCC cell lines, i.e., Huh7, Hep3B, H2M, and PLC-8024 were obtained from the Institute of Virology, Chinese Academy of Medical Sciences (Beijing, China).

**Construction of *RALYL* overexpression and knockdown cells**. To evaluate the function of RALYL, full-length human *RALYL* cDNA was cloned into the pLenti6/V5-TOPO and pLenti6/V5-TOPO-3×FLAGlentiviral expression vector (Invitrogen, Carlsbad, CA). Blasticidin (Sigma-Aldrich, St. Louis, MO) was used to select

for stably transduced cells. For *RALYL* knockdown assay, two short hairpin RNAs (shRNA) specifically targeting RALYL (shRALYL-1, and shRALYL-4) were cloned into the PLL3.7 lentiviral vector (Addgene). Stably transduced cells were selected by puromycin (Sigma-Aldrich).

**Generation of hepatocyte differentiation model**. The derivation of hESCs and their use for research was approved by the ethical committee of the CITIC-Xiangya Reproductive & Genetic Hospital. The chHES-90 cells were established as previously described[39]. In brief, hESC colonies were cultured on human embryonic fibroblasts feeders, which were mitotically inactivated. The hESCs cells were cultured in the medium consisting of knockout DMEM/F12 medium supplemented with 15% knockout serum replacement, 2 mM L-glutamine, 0.1 mM β-mercaptoethanol, 2 mM nonessential amino acids, and 4 ng/mL of basic fibroblast growth factor (Invitrogen). For generation of hepatocyte-like cells, hESCs were passaged onto a feeder free system, and cultured in RPMI-1640 (Life Technologies) supplemented with 100 ng/mL activin A (R&D Systems) and 25 ng/mL Wnt3 a (R&D Systems) for 3 days. To induce hepatic endoderm, cells were grown in KO/DMEM (Life Technologies) supplemented with 25 ng/mL keratinocyte growth factor (R&D Systems) and 2% fetal bovine serum (Gibco) for 2 days, and then further cultured in the KO/DMEM containing 20% SR, 0.1 mM 2-mercaptoethanol, 1 mM glutamine, 1% nonessential amino acids, and 1% dimethyl sulfoxide for 4–7 days. To obtain hepatocyte-like cells, the induced endoderm was cultured in mature medium consisting of 10% FBS, 10 ng/mL hepatocyte growth factor (R&D Systems), 0.5 μM dexamethasone (R&D Systems) and 20 ng/mL oncostatin M (R&D Systems) for 7 more days.

**In vitro functional assays and animals**. Tumorigenic capacities in vitro were evaluated by a cell proliferation kit (Roche Diagnostics, Indianapolis, IN), foci formation assay and colony formation assay in softagar. The spheroid formation assay was performed to assess the self-renewal ability. The migrative and invasive abilities of cells were assessed using transwell migration and invasion assays in vitro. A nude mouse xenograft model was used to evaluate tumor formation and chemoresistant ability in vivo. The liver metastatic ability was assessed by intrasplenic injecting HCC cells into nude mice (Supplementary Materials and "Methods" section). All animal experiments were conducted and approved by the Committee on the Use of Live Animals in Teaching and Research (CULATR) at the University of Hong Kong. The BALB/cAnN-nu (Nude) mice were maintained in a specific pathogen-free animal facility at the University of Hong Kong under 12-h light dark cycles, controlled temperature (~22 °C), and 40–60% humidity with free access to food and water.

**Immunofluorescence**. First, the cells on the chamber slide were fixed with cold methanol and acetone in 1:1 mixture, followed by blocking using 10% FBS for 30 min at 37 °C. After incubation with primary antibodies overnight at 4 °C, the cells were incubated with FITC-conjugated or PE-conjugated secondary antibodies. After counterstaining with DAPI (Roche Diagnostics), the slides were visualized under Carl Zeiss LSM 700 confocal microscope (Carl Zeiss, NY, USA).

**RNA extraction and quantitative real-time PCR (qRT-PCR)**. Total RNA was extracted using the TRIZOL Reagent (Life Technologies). After treating with DNaseI, cDNA was synthesized by reverse transcription (Roche). SYRB Green PCR Kit (Applied Biosystems, Carlsbad, CA) and LightCycler 480 II Real-time PCR Detector (Roche) were used for qRT-PCR analysis. All qRT-PCR reactions were tested in triplicates. Primers used in this study are listed in Supplementary Table 3.

**Western blot analysis**. Quantified protein lysates were resolved on SDS-PAGE, transferred onto a polyvinylidenedifluoride (PVDF) membrane (Millipore), and then blocked with 5% non-fat milk in Tris-buffered saline-Tween 20 (TBS-T) for 1 h at room temperature. The blocked membrane was then incubated with primary antibody diluted 1:1000 in 5% bovine serum albumin in TBS-T at 4 °C overnight.

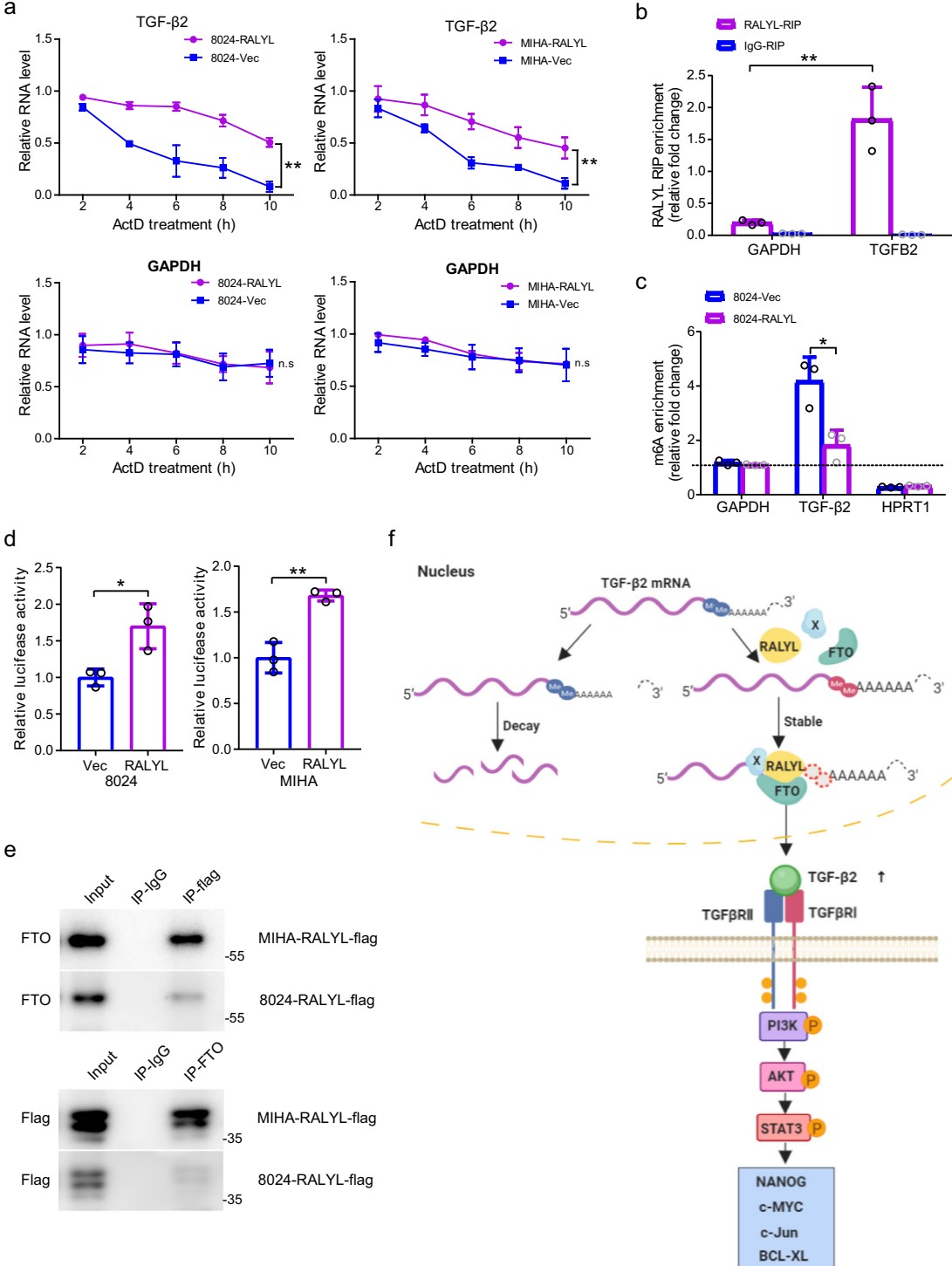

**Fig. 6 *RALYL* upregulates TGF-β2 mRNA stability depending on m⁶A modification. a** PLC-8024 and MIHA cells were transiently transfected with Vec or *RALYL* plasmid for 48 h and then treated with ActD for 0, 2, 4, 6, 8, 10 h. The TGF-β2 mRNA level was determined using qRT-PCR and normalized to 18S rRNA. The GAPDH mRNA level was used as a negative control. The values represent the mean ± SD of 3 independent experiments (*$P < 0.05$, **$P < 0.01$, independent Student's $t$-test). **b** RIP assay using total cell lysates of Huh7 cells was used to assess the interaction between *RALYL* and TGF-β2 mRNA. Enrichment of TGF-β2 mRNA in the *RALYL*-containing immunoprecipitated particles was measured using qRT-PCR and normalized to input. **c** m⁶A RIP and qRT-PCR were used to determine the percentage of TGF-β2 mRNA with m⁶A modification in 8024-Vec and 8024-*RALYL* cells (*$P < 0.05$, Student $t$-test). **d** 3'UTR of TGF-β2 mRNA was fused with firefly luciferase reporter. *RALYL* overexpression augmented the luciferase activity in 8024 and MIHA cells. The values represent the mean ± SD of three independent experiments (*$P < 0.05$, **$P < 0.01$, two-sided Student's $t$-test). **e** The interaction of RALYL with FTO was determined by co-immunoprecipitation with anti-FTO (IP: FTO) and anti-FLAG antibody (IP: FLAG) or IgG (IP: IgG) in *RALYL*-flag-transfected cells. Total cell lysate (Input) was used as a positive control. **f** A schematic diagram illustrating the proposed *RALYL* enhanced stemness-related features in HCC. RALYL could decrease m⁶A modification of TGF-β2 mRNA, thereby maintain its stability by interaction with FTO. By sustaining the secretion of TGF-β2, the PI3K/AKT and STAT3 were activated to increase the stemness of HCC.

All antibodies used are listed in Supplementary Table 4. After washing with TBS-T, the membrane was incubated for 1 h with horseradish peroxidase (HRP)-conjugated secondary antibody. A complex of primary and secondary antibodies-labeled proteins were detected by enhanced chemiluminescence (ECL) system followed by exposure to Amersham Imager 600 (GE Healthcare).

**Cell sorting**. PE-conjugated anti-human CD133 antibody (MiltenyiBiotec, Ber-gischGladbach, Germany) was used for cell sorting. The PE-conjugated isotype mouse immunoglobulin G1b (MiltenyiBiotec) was used as the control. For positively stained cells, only the top 15% most brightly stained were selected as CD133-positive populations. Meanwhile, the bottom 15% most dimly stained cells were selected as CD133-negative cells. Samples were sorted using the FACS Aria I Cell Sorter (BD Biosciences).

**Chemotherapy-induced cytotoxicity and apoptotic assay**. Chemotherapy-induced cytotoxicity (CDDP, 5-FU, Sorafenib) was determined by XTT Cell Proliferation Assay (Roche Diagnostics) according to the manufacturer's instructions. The apoptotic assay was determined by flow cytometry. After treating with CDDP or 5-FU for 48 h, the cells were collected and double stained with FITC-conjugated Annexin-V and PI provided in the BD apoptosis detection kit (BD Biosciences). All results are expressed as mean ± SD of three independent repeats. The analysis was performed using the FACS Canto II Analyzer (BD Biosciences) and FlowJo software (Tree Star).

**RNA stability assay**. PLC-8024 and MIHA were transfected with *RALYL* and control plasmids. *RALYL* was highly expressed in PLC-8024 and MIHA 48 h after transfection. Thereafter, those cells were treated with Actinomycin D (Sigma-Aldrich) at 5 μg/mL. The time courses of samples with Actinomycin D treatment (0, 2, 4, 6, 8, and 10 h) were used for RNA extraction. RNA was reversed transcription and analyzed by quantitative real-time PCR (qRT-PCR).

**RNA immunoprecipitation (RIP)**. The RIP-Assay Kit (MBL, Japan) was used for pulling down mRNA complexes according to the manuscript's instruction. Briefly, $5 \times 10^7$ cells in 15-cm dish were harvested and lysed in lysis buffer. After pre-clearing with 25 μL of protein A agarose beads (GE Healthcare) for 1 h at 4 °C, 10 μL of the supernatant was saved as input, and the remaining supernatant was incubated overnight at 4 °C with protein A agarose beads coated with either 15-μg anti-*RALYL* (Sigma-Aldrich) antibody or normal rabbit IgG polyclonal antibody (MBL). Next, RNA was isolated from the antibody-immobilized protein A agarose beads-RNP complex. RNAs enriched in the RNP complex were analyzed using qRT-PCR.

**Gene-specific m⁶A qPCR**. An antibody recognized $m^6A$ was used for RIP using MBL RIP-Assay Kit as described in the RNA immunoprecipitation section. qRT-PCR was performed to evaluate the relative abundance of specific mRNA in m6A RIP complexes, and input samples between the *RALYL*-overexpression and control cell lines. The HPRT1 mRNA without $m^6A$ modification was chosen as internal control.

**Luciferase reporter assay**. pmirGlo luciferase expression system was purchased from Promega Corporation (Australia) and performed according to the manufacturer's instructions. TGF-β2 reporter plasmid was cloned by inserting the full-length TGF-β2-3′UTR after the Firefly luciferase (F-luc) coding sequence. Cells seeded in 96-well plate were transfected with 100 ng of F-luc-TGF-β2-3′UTR fusion reporter plasmid. After 72 h, cells were analyzed with Dual-Glo Luciferase system (Promega). Firefly luciferase (F-luc) activity was used to evaluate the effect of m⁶A modification on TGF-β2-3′UTR. Renilla Luciferase (R-luc) was used to normalize the transfection efficiency of the reporter plasmid.

**Statistical analysis**. The SPSS version 17.0 (SPSS, Inc., Chicago, IL) was used for data analysis. Patients' survival rates were analyzed using Kaplan–Meier plots and log-rank tests. The correlations between *RALYL*, CD133, TGF-β2, and different clinicopathological parameters were evaluated using Pearson's $\chi^2$ test. Univariable and multivariable Cox proportional hazard regression models were used to analyze independent prognostic factors. The frequency of CSCs with tumor initiation capabilities was calculated using limiting dilution assay in the ELDA software[40]. Data are presented as the mean ± SD of three independent experiments. Results were considered statistically significant for *P* values <0.05.

**Reporting summary**. Further information on research design is available in the Nature Research Reporting Summary linked to this article.

## Data availability
All data supporting the findings of this study are available within the paper and its supplementary information files. The raw and processed sequencing data are available in Gene Expression Omnibus under accession GSE163601. Source data are provided with this paper. All relevant data are available from the authors on reasonable request.

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

## Acknowledgements

This work was supported by grants from the Hong Kong Research Grant Council (RGC) including GRF (17143716 and 767313), Collaborative Research Funds (C7065-18GF and C7026-18GF), Theme-based Research Scheme (T12-704/16-R), National Natural Science Foundation of China (81772554 and 81802316), Shenzhen Science and Technology programs (KQTD2018041118502879 and KQDT2015033117210153), Shenzhen Fundamental Research Program (JCYJ20180508153249223), Shanghai Municipal Education Commission-Gaofeng Clinical Medicine Grant Support (RC20200037), Shanghai Rising-Star Program grant(KY2019460). Professor XY Guan is Sophie YM Chan Professor in Cancer Research. Professor IOL Ng is Loke Yew professor in Pathology.

## Author contributions

X.W., L.J., and X.Y.G initiated and designed the experiments; X.W., J.W., C.S., Y.W., Q.Y., M.C., and C.J. performed the experiments; X.W. analyzed the data and interpreted the data; X.W., J.W., L.J., and Z.F. performed statistic analyses; Y.F.Y. provided the HCC clinical samples and the relevant clinical information; I.O.L.N provided mouse liver organoids. Y.M.T. and X.Z. performed mouse liver organoid related experiments. C.M.W., M.L., H.C., and I.O.L.N provided valuable comments; X.W. draft the manuscript; L.J. and X.Y.G revised the manuscript and supervised the study.

## Competing interests

The authors declare no competing interests.
