## [Peer Review File · Nature Communications]

Reviewers' Comments:

Reviewer #1:

Remarks to the Author:

In the here presented manuscript, Wang et al. performed multiple analyses to identify and validate potential regulators of cancer stem cell features in hepatocellular carcinoma (HCC). By performing molecular analyses on ES-induced stages of the hepatocyte lineage, the authors identified RALYL as a candidate gene to be associated with adverse biological features including poor overall survival of patients and activation of pro-oncogenic and pro-metastatic features in HCC. Functional and in vivo studies confirmed activation of stemness features upon positive RALYL regulation, which was a consequence of sustained TGF- β 2 mRNA stability. The authors obtained complex data from multiple approach including different molecular, in vitro, and in vivo analyses.

Key findings:

- Expression of RALYL is positively associated with expression of stemness markers, namely CD133, AFP, NANOG, SOX2 and patients with high expression exhibit worst overall survival and disease-free survival
- High expression of RALYL enhances tumor cell proliferation, tumorigenicity, as well as migration and invasion.
- RALYL expression promotes resistance to common chemotherapeutic agents, which is reflected in reduced apoptosis in vitro, and bigger tumor size in vivo.
- Cancer stemness is regulated through RALYL by STAT3-dependant upregulation of TGF- β 2 regulated. RALYL stabilizes the mRNA of TGF- β 2 by decreasing m6A modifications, which further allows upregulation of TGF- β 2.

The study addresses an important aspect of cancer stemness in primary liver cancer, and it is technically well performed. The manuscript is well structured and written. However, potential translational implications should be validated in independent patient cohorts. Furthermore, mechanistic analyses should be extended, particularly related to the role of RALYL as bona fide oncogene and the association to TGF- β 2.

Major comments:

♣ Adjacent liver tissue has very similar expression signature like HCC tissue. To conclusively judge the reliability of the model isolated hepatocytes and non-diseased livers should be included in Figure 1. Adjacent tumor tissue seems not suitable as a normal control given the altered microenvironment and mix of different cell types. Furthermore, selection of the markers should be explained in more detail. Is this a pre-selected list of genes (reference) or were these genes identified by unsupervised methods? Statistical pipeline should be delineated (Figure 1B).

• Expression level of CSC and stemness related markers (CD133, CD24, C-MYC) should have also been demonstrated at different stages of hepatocyte differentiation, not just for established hepatoma and immortalized hepatocytes cell lines (Supplementary Figure 1B and C). Result should support the finding that high expression of RALYL in LP and HL is followed by increased expression of these markers, particularly CD133.

Authors should also more clearly delineate how and why RALYL gene was selected as a potential candidate gene with implications for cancer stemness. This is unclear as presented.

• Univariate analyses of 117 HCC samples showed that patients with higher RALYL expression positively correlated with poorer differentiation state, vascular invasion, metastases, as well as poor overall survival. This finding should further be supported with the data regarding CD133 expression in these tumors as well as TGF- β 2 status. Furthermore, in concordance with REMARK guidelines, multivariate analyses should be performed to confirm independent prognostic implications. In line with this, results should be confirmed in an independent cohort of authentic HCC, e.g. publically available datasets.

• Tumorigenicity of two "normal" (L02 and MIHA) hepatocyte cell lines after forced expression of RALYL is potentially interesting since this would establish RALYL as a bona fide oncogene. However, this requires more thorough investigations and should be extended (e.g. induction in isolated hepatocytes). In addition, previous studies showed that both MIHA and L02 do not express CD133 (PMID 17570225). This should be explained. In addition, both Huh7 (mutation)

and Hep3B (null) show alterations in p53. What is the status in LO2 and MIHA? It would be interesting to include e.g. HepG2 as p53 WT cell line to exclude p53-dependent mechanisms.

- Difference in proliferation upon transfection with RALYL is not striking. Significant difference is present only at some time points (Figure 2B and Supplementary Figure 2B).
- Western blot results, particularly for overexpression, are not of sufficient quality and unconvincing as presented (SLUG, Fibronectin, E-Cadherin). Quality should be improved (Figure 3D). Also applies to Figure 5B – some western blots are of insufficient quality and difficult to interpret. This should be improved.
- The authors should explain the selection of the utilized chemotherapeutic agents. It would be important to explore approved first line therapies for HCC (i.e. sorafenib, lenvatinib). This should be included.
- Mechanistic investigations, in particular the role of TGF- β 2 are potentially interesting and relevant. However, majority of results were obtained by Hep3B, Huh7, MIHA and LO2. It is unclear why 8024 was used for the transcriptome analyses. At least one more cell line should be included in the analyses. It would also be important demonstrate effect on target genes after silencing of RALYL expression.

Minor comments

- ♣ Supplementary Figure 1 – correct figure legend (Four instead of three cell lines).
- ♣ Axis on flow cytometry dot plots should be adequately labeled (Supplementary Figure 4E)
- ♣ Intrasplenic tumor injections are frequently limited by tumor growth within the spleen (Andersen et al. Sci Transl. Med 2010). Please comment.

Reviewer #3:

Remarks to the Author:

This work used an in vitro hepatocyte differentiation system to identify molecular signatures of liver progenitor cells and premature hepatocytes that may contribute to the 'stemness' of hepatocellular carcinoma cells. Transcriptomic profiling identified RALYL as a factor that could upregulate other stemness-related markers. This work revealed that increased RALYL expression upregulates stem-ness factors, results in larger tumors in a mouse xenograft model, promotes HCC migration and invasion, increases resistance to common chemotherapeutics, and correlates with poor prognosis in patient cohorts. Upregulation of RALYL results in increased TGF- β signaling, which in turn regulates pathways that regulate cell 'stem-ness'. RALYL expression stabilizes TGF- β mRNA, which the authors postulate may be due to the loss of m6A methylation. Overall I think this work is solid and of interest, and I do not see major experiments that would need to be included for publication. I do, however, think that the conclusion that m6A is involved is premature (explained below), so I would suggest either removing those data or performing additional experiments to validate this conclusion. Some points that should be addressed prior to publication are listed below.

Significant points:

1. It is a little bit unclear whether the differentiation model described here is new to this report or not. In the main text it is referred to as new, but in the methods a paper in press is referenced. This may just be a consequence of delays and multiple papers being under review at the same time, but this should be clarified in the revised manuscript.
2. The poor quality of many of the western blots in figure 3D make it difficult to assess the authors claims about EMT markers. The results in 3A-C support the claim that RALYL induces migration and invasion, so perhaps this could be removed (and the text adjusted) if alternative expression data is not available.
3. Please clarify what is meant by the statement that "a rare expression change was detected in the control cells (Figure 4D and Supplementary Figure 4C)..." (it would be helpful to specifically indicate which lanes we should be comparing).
4. While there does, indeed, seem to be a difference in stability in TGF β mRNA when RALYL is

overexpressed, the model that this stability is regulated through m6A would require more experimental validation than the m6A-IP that is presented (particularly given the fact that m6A antibodies have been shown to recognize related modifications). Since this is a relatively minor point in the paper, I would suggest removing the m6A data. If the authors wish to pursue this further, they would need additional experiments to validate the involvement of m6A, such as testing the effects of m6A-regulatory proteins (METTL3/METTL14, YTHDF2 etc), and/or more detailed mapping of putative m6A sites. The authors should also consider other possible mechanisms of mRNA stability regulation.

Minor points:

1. Labeling is not always totally clear. For instance, the heat map in 1B has the HL abbreviation, but this doesn't seem to be defined anywhere.
2. The 'RNA stability' methods section may need to be reworded a bit – as written it appears that the Actinomycin D treatment was for 48 hours when I believe the authors meant to state that the time course of samples (0 hr -> 10 hrs) was collected after cells had been transfected for 48 hours.
3. Some of the figures are presented in a confusing format. For instance, in supplementary figure 1A-C, qPCR data are presented as connected lines as if to indicate a continuous variable on the x axis. While the argument could be made that the progression from ES \diamond HEP does represent a time-like variable, this is definitely not the case in panel 1C where the x-axis simply lists different target genes.

A point-by-point response to the Reviewers' comments and suggestions

Reviewer #1

The study addresses an important aspect of cancer stemness in primary liver cancer, and it is technically well performed. The manuscript is well structured and written. However, potential translational implications should be validated in independent patient cohorts. Furthermore, mechanistic analyses should be extended, particularly related to the role of RALYL as bona fide oncogene and the association to TGF- β 2.

Major comments:

1. Adjacent liver tissue has very similar expression signature like HCC tissue. To conclusively judge the reliability of the model isolated hepatocytes and non-diseased livers should be included in Figure 1. Adjacent tumor tissue seems not suitable as a normal control given the altered microenvironment and mix of different cell types. Furthermore, selection of the markers should be explained in more detail. Is this a pre-selected list of genes (reference) or were these genes identified by unsupervised methods? Statistical pipeline should be delineated (Figure 1B).

Our reply:

Thanks to the reviewer's comments. We agree with the reviewer's suggestion that non-diseased liver is more appropriate, but it is difficult for us to collect sample and perform the transcriptome sequencing during the current situation of Covid-19 pandemic. We will use healthy liver in our further study.

According to literature, the representative specific markers are selected for each liver development stage (Aaron M. Zorn, liver development, StemBook, 2008). The pre-selected list of genes was used to investigate whether the *in vitro* liver development model used in our study is reliable. The RNA-seq results indicated that most of the markers are indeed highly expressed in their corresponding stage, therefore, we believed the *in vitro* liver development model is reliable. As the reviewer raised this issue, it seems a bit confusing to show too many markers in Figure 1b. Therefore, the number of markers has been narrowed down to 3-4 genes for each stage (see revised Figure 1b).

2. Expression level of CSC and stemness related markers (CD133, CD24, C-MYC) should have also been demonstrated at different stages of hepatocyte differentiation, not just for established hepatoma and immortalized hepatocytes cell lines

(Supplementary Figure 1B and C). Result should support the finding that high expression of RALYL in LP and HL is followed by increased expression of these markers, particularly CD133.

Authors should also more clearly delineate how and why RALYL gene was selected as a potential candidate gene with implications for cancer stemness. This is unclear as presented.

Our reply:

According to Reviewer's suggestion, expression pattern of HCC stemness related markers CD133, C-MYC and EpCAM were tested by qRT-PCR in our *in vitro* liver development model. The results showed that CD133 and EpCAM were highly expressed in LP and PH as RALYL (see revised Supplementary Figure 1a). Furthermore, the expressions of CD133, C-MYC and EpCAM were also increased in RALYL-transfected PLC-8024 and MIHA cells (see revised Supplementary Figure 1c). This information has been added to the revised "Results" section (the first paragraph, page #5).

Sorry for the unclear statement in our original manuscript about how and why RALYL gene was selected. In the present study, we aim to identify and characterize genes related to the stemness maintenance and regulation. The selection priority includes gene(s) encoding nuclear protein which show similar expression pattern to liver progenitor markers. Several genes including RALYL were selected and transfected into immortalized liver cell line MIHA and HCC cell line PLC-8024. Spheroid formation assay was used to assess the self-renew ability of those candidate genes, and RALYL showed strongest spheroid formation ability. In addition, we found that RALYL showed strongest ability to upregulate stemness markers, such as CD133, AFP, C-MYC and NANOG. Therefore, RALYL was selected for further study. This information has been added to the revised "Results" section (the first paragraph, page #5).

3. Univariate analyses of 117 HCC samples showed that patients with higher RALYL expression positively correlated with poorer differentiation state, vascular invasion, metastases, as well as poor overall survival. This finding should further be supported with the data regarding CD133 expression in these tumors as well as TGF- β 2 status.

Furthermore, in concordance with REMARK guidelines, multivariate analyses should be performed to confirm independent prognostic implications. In line with this, results should be confirmed in an independent cohort of authentic HCC, e.g. publically available datasets.

Our reply:

In accordance with reviewer's suggestion, we also assessed the expression of CD133 and TGF- β 2 in the 117 HCC cases. Association study found that high CD133 expression was significantly associated with poor differentiation stage ($P=0.034$) and metastasis ($P=0.050$), and high expression of TGF- β 2 was positively correlated with poor differentiation stage ($P=0.009$), vascular invasion ($P=0.014$) and metastasis ($P=0.024$) (Table 2 and Table 3). As the expression of RALYL was positively correlated with poorer differentiation state, vascular invasion and metastases, there were quite consistent among these genes in clinical significance in our cohort. In addition, the high expression of RALYL was significantly associated with higher expression of CD133 ($P=0.003$) and TGF- β 2 ($P=0.029$) (revised Table 1). Cox proportional hazard regression analysis showed that RALYL expression was an independent prognostic factor for overall survival of HCC patients ($P=0.046$; Table 4). In terms of survival, the expression CD133 was closely associated with poor prognosis in our in-house cohort ($P=0.021$) as well as TCGA database ($P=0.004$). Similarly, higher expression of TGF- β 2 is associated with poorer overall survival rate in both our in-house cohort ($P=0.002$) as well as TCGA database ($P=0.047$) (revised Supplementary Figure 1f). In line with the prognostic implications of RALYL in 117 patient samples, high expression of RALYL is closely associated with poor overall survival rate ($P=0.006$) in TCGA database (revised Figure 1f). All these results indicate that RALYL, CD133 and TGF- β 2 have similar clinical implications. This information has been added into the revised 'Results' and 'Discussion' section (The second paragraph, Page #6; the first paragraph, Page #14).

4. Tumorigenicity of two "normal" (LO2 and MiHA) hepatocyte cell lines after forced expression of RALYL is potentially interesting since this would establish RALYL as a bone fide oncogene. However, this requires more thorough investigations and should be extended (e.g. induction in isolated hepatocytes). In addition, previous studies showed that both MiHA and LO2 do not express CD133 (PMID 17570225). This should be explained. In addition, both Huh7 (mutation) and Hep3B (null) show

alterations in p53. What is the status in LO2 and MIHA? It would be interesting to include e.g. HepG2 as p53 WT cell line to exclude p53-dependent mechanisms.

Our reply:

We agree with reviewer's points that the function roles of RALYL as a bona fide oncogene should be extended in isolated hepatocytes. We have endeavored to isolate fresh hepatocytes in adjacent non-tumor liver tissues and transfect RALYL into the hepatocytes. Unfortunately, it is really difficult to do the transfection and passage the fresh hepatocytes to perform *in vitro* functional studies.

Yes, according to our previous work (Ma et al., Gastroenterology, 2007.), CD133 is hardly detected in MIHA and LO2. Because of the gate setting in flow cytometry, we acquired higher percentage of CD133⁺ cells in the original manuscript. By resetting the gate in accord with our previous paper, the percentage of CD133⁺ cells in MIHA was similar in our two works (see revised Figure 1d).

MIHA and LO2 possess wild-type TP53 (Yamamoto et al., Cancers, 2019. Li et al., BMC cancer, 2017.), therefore, molecular biology experiments performed in those two cell lines could exclude the p53-dependent mechanism.

5. Difference in proliferation upon transfection with RALYL is not striking. Significant difference is present only at some time points (Figure 2B and Supplementary Figure 2B).

Our reply:

Thanks for the reviewers' comments. Two more independent experiments for each cell line were performed with three repeats. In revised manuscript, all data of three independent experiments with error bar has been presented. Although it is not striking at some time points, difference between experiment and control group is significant at statistical level ($P < 0.05$). The P values were shown in revised Figure 2B and Supplementary Figure 2B.

6. Western blot results, particularly for overexpression, are not of sufficient quality and unconvincing as presented (SLUG, Fibronectin, E-Cadherin). Quality should be improved (Figure 3D). Also applies to Figure 5B – some western blots are of insufficient quality and difficult to interpret. This should be improved.

Our reply:

Thanks for the comments. We have repeated western blotting. Western blotting images with high quality replaced the original ones in revised Figure 3D and Figure 5B.

7. The authors should explain the selection of the utilized chemotherapeutic agents. It would be important to explore approved first line therapies for HCC (i.e. sorafenib, lenvatinib). This should be included.

Our reply:

Hepatic arterial infusion chemotherapy (HAIC) with 5-fluorouracil (5-FU) and cisplatin (CDDP) is commonly used for advanced hepatocellular carcinoma (HCC). (Park et al., Cancer, 2007. Ueshima et al., Oncology, 2010.) Thus 5-Fu and CDDP were used as chemotherapeutic agents in our present study. Currently, sorafenib and lenvatinib are the approved first line therapy for HCC. In the present study, we found that RALYL could increase drug resistance for 5-Fu and CDDP treatment, but not for sorafenib or lenvatinib (see revised supplement Figure 4c for sorafenib results, lenvatinib results were not shown).

8. Mechanistic investigations, in particular the role of TGF- β 2 are potentially interesting and relevant. However, majority of results were obtained by Hep3B, Huh7, MIHA and LO2. It is unclear why 8024 was used for the transcriptome analyses. At least one more cell line should be included in the analyses. It would also be important demonstrate effect on target genes after silencing of RALYL expression.

Our reply:

MIHA, LO2 and PLC-8024 were used for RALYL overexpressing system, while Huh7, Hep3B and H2M were used for RALYL knockdown experiments. PLC-8024 was used for all functional and molecular experiments in our study (Figure 1, 2, 3, 4, 5 and 6, Supplementary Figure 1, 4 and 5). At first, PLC-8024 was used for the transcriptome analyses to unveil the mechanism underlying RALYL over-expression. It was found that TGF- β 2 as well as several genes was up-regulated in RALYL over-expressing cells. Thereafter, qRT-PCR was performed to detect the expression level of TGF- β 2 and other genes in Hep3B, Huh7, MIHA and LO2. The up-regulation of TGF- β 2 can be observed in PLC-8024, MIHA, LO2 with RALYL overexpression, consistently, the expression level of TGF- β 2 was decreased in RALYL knockdown Huh7, Hep3B and H2M cells (See in revised Supplementary Figure 5C). As qRT-

PCR results are consistent in all HCC cells, transcriptome analysis of more HCC cells was not carried out. Due to the Covid-19 pandemic, it's also hard for us to perform transcriptome sequencing for other cell lines.

Minor comments

1. Supplementary Figure 1 – correct figure legend (Four instead of three cell lines).

Our reply:

Thanks! We have corrected it accordingly. (See in Supplementary Figure 1)

2. Axis on flow cytometry dot plots should be adequately labeled (Supplementary Figure 4E)

Our reply:

The Axis on flow cytometry in Supplementary Figure 4E has been adequately labeled (See in Supplementary Figure 4e).

3. Intrasplenic tumor injections are frequently limited by tumor growth within the spleen (Andersen et al. Sci Transl. Med 2010). Please comment.PMID: 20962331

Our reply:

We understand the Reviewer's concern. Intrasplenic injection is a widely used as liver metastasis model (Endo et al., Cancer Gene Therapy, 2002.). Some studies indicated that the metastasis in liver of nude mouse did not affected by the tumor formation in the spleens (Giavazzi et al., Cancer Research, 1986.). In our study, visible tumor formation in spleen was not commonly observed and only detected in 4/27 of mice intrasplenic injected with different cells. Therefore, we believe that being a HCC metastatic mouse model, intrasplenic injection is better than tail vein injection which often causes pulmonary metastasis (Kozlowski et al, Cancer Research, 1984).

Reviewer #3 (Remarks to the Author):

Overall I think this work is solid and of interest, and I do not see major experiments that would need to be included for publication. I do, however, think that the conclusion that m6A is involved is premature (explained below), so I would suggest either removing those data or performing additional experiments to validate this conclusion. Some points that should be addressed prior to publication are listed below.

Significant points:

1. It is a little bit unclear whether the differentiation model described here is new to this report or not. In the main text it is referred to as new, but in the methods a paper in press is referenced. This may just be a consequence of delays and multiple papers being under review at the same time, but this should be clarified in the revised manuscript.

Our reply:

The liver differentiation model has been studied by RNA-seq in another paper (Ming Liu et al., PNAS, 2020). In that paper, we described how to establish the *in vitro* liver differentiation model and how to discover two subtypes of liver cancer with different oncofetal properties and therapeutic targets using that model. When we submitted this manuscript to *Nature Communications*, that work was under revision. In this work, RALYL was selected as a stemness-related gene for further study.

2. The poor quality of many of the western blots in figure 3D make it difficult to assess the authors claims about EMT markers. The results in 3A-C support the claim that RALYL induces migration and invasion, so perhaps this could be removed (and the text adjusted) if alternative expression data is not available.

Our reply:

Sorry for the poor quality of western blotting images. According to Reviewer's suggestion, we have repeated the western blotting. Images were replaced with high quality ones in revised Figure 3D.

3. Please clarify what is meant by the statement that "a rare expression change was detected in the control cells (Figure 4D and Supplementary Figure 4C)." (it would be helpful to specifically indicate which lanes we should be comparing).

Our reply:

Sorry for the confusing statement in the original manuscript. It is indeed difficult to distinguish the difference between atRA treatment and no treatment groups by conventional PCR results. Therefore, qRT-PCR was performed to compare the fold change of the mature hepatocyte markers and the stemness markers between RALYL overexpression and control groups. We found that the fold changes of expressions of mature hepatocyte markers (CK8, CK18, and albumin) were significantly higher in

8024-RALYL and MIHA-RALYL cell after atRA treatment, compared with 8024-Vec and MIHA-Vec. Whereas, the fold change of expressions of stemness-associated genes (AFP, CD133, and NANOG) and RALYL were significantly lower in 8024-RALYL and MIHA-RALYL cells compared with controls (Figure 4d), suggesting that RALYL-expressing cells possessed higher differentiation potential than controls. The qRT-PCR results were shown in revised Figure 4D.

4. While there does, indeed, seem to be a difference in stability in TGF β mRNA when RALYL is overexpressed, the model that this stability is regulated through m⁶A would require more experimental validation than the m⁶A-IP that is presented (particularly given the fact that m⁶A antibodies have been shown to recognize related modifications). Since this is a relatively minor point in the paper, I would suggest removing the m⁶A data. If the authors wish to pursue this further, they would need additional experiments to validate the involvement of m⁶A, such as testing the effects of m⁶A-regulatory proteins (METTL3/METTL14, YTHDF2 etc), and/or more detailed mapping of putative m⁶A sites. The authors should also consider other possible mechanisms of mRNA stability regulation.

Our reply:

Thanks to the reviewer for the comment. According to Reviewer's suggestion, additional experiments were performed to validate the m⁶A modification by RALYL. As is known, m⁶A modification usually presents in the consensus sequence RRACH and the enrichment of m⁶A in 3' untranslated regions (3' UTRs) near stop codon is reported to regulate the stability of mRNA (Zhao et al., Molecular Cell Biology, 2017). We found that multiple RRACH sites were located in 3' UTR of TGF- β 2 mRNA around stop codon. The luciferase reporter assays confirmed the interaction between RALYL and 3'-UTR of TGF- β 2 mRNA as the relative luciferase activity in RALYL-overexpressing cells was significantly augmented compared with control cells. Furthermore, we investigated whether RALYL could cooperative two well-known m⁶A erasers, including Fat mass and obesity-associated protein (FTO) and alkB homologue 5 (ALKBH5) and the m⁶A reader YTHDF2. IP assays are performed to confirm the interaction between RALYL and several well-known m⁶A erasers in exogenously expressed stable 8024-RALYL-flag and MIHA-RALYL-flag cells. It was excited to find that FTO is pulled by flag-RALYL, but ALKBH5 and YTHDF2

were not detected in immunoprecipitation. Thus, we concluded that RALYL could cooperate with FTO to remove m⁶A of TGF-β2 mRNA and keep its stability. All those results were added to revised manuscript (The first paragraph, Page #16).

Minor points:

1. Labeling is not always totally clear. For instance, the heat map in 1B has the HL abbreviation, but this doesn't seem to be defined anywhere.

Our reply:

Sorry for the language errors. We have revised the abbreviations in Figure 1b. The full names for induced human embryonic stem (ES) cells, definitive endoderm (DE), liver progenitor cells (LP) and premature hepatocytes (PH) were added in revised figure legend.

2. The 'RNA stability' methods section may need to be reworded a bit – as written it appears that the Actinomycin D treatment was for 48 hours when I believe the authors meant to state that the time course of samples (0 hr -> 10 hrs) was collected after cells had been transfected for 48 hours.

Our reply:

Thank you for the reviewer's comments. We have rewritten the 'RNA stability' methods section. The revised section was seen in the first paragraph, Page #20.

3. Some of the figures are presented in a confusing format. For instance, in supplementary figure 1A-C, qPCR data are presented as connected lines as if to indicate a continuous variable on the x axis. While the argument could be made that the progression from ES HEP does represent a time-like variable, this is definitely not the case in panel 1C where the x-axis simply lists different target genes.

Our reply:

Thanks for reviewer's kind reminder. Yes, we totally agree with the reviewer's suggestions. We have replaced the line chart by bar chart to show qPCR data (see in revised supplementary figure 1A-C).

Reviewers' Comments:

Reviewer #1:

Remarks to the Author:

This is the revised version of the manuscript by Wang et al.

The authors experimentally addressed several of the raised concerns and clarified several of the comments. The revision strengthened the study. However, several other major concerns have only been partly or not addressed by the authors. In particular, translational implications are only completely delineated. The study still misses non-diseased liver tissue as controls for the study. The provided answer related to the COVID19 situation does not suffice. While the impact of the pandemic for translational science is enormous and certainly appreciated, archived tissue can be utilized or the authors could establish collaborations to address this critical issue. In addition, preliminary investigations using publically available data would be a viable option. While it is still unclear why the selection of genes was reduced to only few genes despite availability of whole transcriptome data, the reviewer appreciates that the graphical representation in Figure 1 is now quite intuitive.

The prognostic implications are still not quite clear. How were the variables for the multivariate analyses selected? Why are essential clinico-pathological associations from Table 1 as well as CD133 and TGFB2 not included? Furthermore, it is unclear how the TCGA data was retrieved and analyzed. Validation of the findings by GEPIA2 could not confirm the presented findings for CD133, RALYL and TGFB2. Please explain and provide the pipeline of the analyses. Of note, it would be desirable that the sequencing data is deposited in a publically accessible database and made available not only on "reasonable request". It is unclear why the authors selected a different gating strategy for CD133. This might dilute the fraction of putative CSCs and, thus, not be meaningful. The lack of validation of the findings in freshly isolated hepatocytes remains a significant shortcoming that could be overcome, e.g. using organoid cultures.

Reviewer #3:

Remarks to the Author:

This work demonstrates that RALYL is a critical regulator of HCC cancer stem cell populations via its effects on TGF-B2 mRNA stability and PI3K/AKT and STAT3 signaling. The authors also present intriguing data suggesting that this effect on mRNA stability could be mediated via N6-methyladenosine in the TGF-B2 3'UTR. My primary concerns with the initial submission were that I could not interpret the western blots through which the authors aimed to demonstrate changes in markers of EMT upon overexpression or knockdown of RALYL, and that there was insufficient evidence to support the model that the observed change in RALYL stability is regulated by N6-methyladenosine (m6A). With respect to the first point, the authors have repeated the western blots so that they can now be interpreted much more easily. They have also added a few functional experiments testing the possibility that m6A may be involved, including assays that suggest interactions between RALYL and TGF-B2, and between RALYL and the m6a demethylase, FTO. In addition, they have addressed the minor points that I brought up, and I think the manuscript is appropriate for publication in its current revised form.

Reviewer #1 (Remarks to the Author):

This is the revised version of the manuscript by Wang et al.

The authors experimentally addressed several of the raised concerns and clarified several of the comments. The revision strengthened the study. However, several other major concerns have only been partly or not addressed by the authors. In particular, translational implications are only completely delineated.

Comments 1:

The study still misses non-diseased liver tissue as controls for the study. The provided answer related to the COVID19 situation does not suffice. While the impact of the pandemic for translational science is enormous and certainly appreciated, archived tissue can be utilized or the authors could establish collaborations to address this critical issue. In addition, preliminary investigations using publically available data would be a viable option.

Our reply:

According to Reviewer’s suggestion, RNA-seq was performed with two normal liver specimens. In general, the expression profiles are quite similar between normal liver (NL) and hepatocytes (HEP) adjacent to HCC tissues, especially for those selected genes in our original manuscript (Fig. 1b), which is shown in this letter as Figure 1. qRT-PCR was also applied to validate the expression level of some well-known markers of each stage and results showed similar expression level of these markers in HEP and NL (original Supplementary Fig. 1a, Figure 2 in this letter). In the revised Fig. 1b and Supplementary Fig 1a, selected genes in HEPs were replaced by NLs (see revised Fig. 1b, revised Supplementary Fig. 1a).

Figure 1

Figure 2

Comments 2:

While it is still unclear why the selection of genes was reduced to only few genes despite availability of whole transcriptome data, the reviewer appreciates that the graphical representation in Figure 1 is now quite intuitive.

Our reply:

The purpose of this study is to identify and characterize genes related to the stemness maintenance and regulation in HCC cancer stem cells (CSCs) from our recently established *in vitro* hepatocyte differentiation model. As described in the manuscript, this model includes embryonic stem (ES) cells, definitive endoderm (DE), liver progenitor (LP) cells and premature hepatocytes (PH). RNA-seq was performed with these four stage cells, as well as two pairs of HCC clinical specimens (HEP and HCC). A total of 20,038 protein-coding genes were detected by RNA-seq. 2,324 genes were selected for those highly expressed in LP cells compared to ES, EN, PH cells and clinical samples. Next, 126 genes which expression level are much higher in HCC cells (> 5 fold) compared with non-tumor cells were selected. 34 genes encoding nuclear proteins were then selected. Based on literature review, 9 not-well studied genes were selected for further study. Spheroid formation assay was used to study the self-renew ability of these candidate genes in immortalized liver cell line MIHA and HCC cell line PLC-8024, and RALYL showed strongest spheroid formation ability. In addition, RALYL also showed the strongest ability to up-regulate stemness markers, such as CD133, AFP, C-MYC and NANOG. Therefore, RALYL was further investigated in the present study.

Comments 3:

The prognostic implications are still not quite clear. How were the variables for the multivariate analyses selected? Why are essential clinico-pathological associations from Table 1 as well as CD133 and TGFB2 not included?

Our reply:

According to Reviewer's suggestion, we included more related variables such as 'cirrhosis', 'vascular invasion', 'CD133' and 'TGFB2' expression to the multivariate analyses. Results showed that RALYL is an independent prognostic indicator for the

survival of HCC patients. This result has been added in the revised manuscript (revised Table 2).

Comments 4:

Furthermore, it is unclear how the TCGA data was retrieved and analyzed. Validation of the findings by GEPIA2 could not confirm the presented findings for CD133, RALYL and TGFB2. Please explain and provide the pipeline of the analyses.

Our reply:

The process of the TCGA data retrieval and analysis are as follows:

1) Data retrieval: The data of the TCGA dataset is freely accessible. Efforts by centers (e.g. the University of California Santa Cruz) allow users to easily access the normalized TCGA RNA-seq, methylation and clinical data in excel files. We retrieved and downloaded normalized RNA-seq data and corresponding clinical information of the Liver Cancer (LIHC) cohort from the TCGA hub in UCSC XENA website ([https://xenabrowser.net/datapages/?cohort=TCGA%20Liver%20Cancer%20\(LIHC\)&removeHub=https%3A%2F%2Fxcena.treehouse.gi.ucsc.edu%3A443](https://xenabrowser.net/datapages/?cohort=TCGA%20Liver%20Cancer%20(LIHC)&removeHub=https%3A%2F%2Fxcena.treehouse.gi.ucsc.edu%3A443)).

2) Data analysis: After screening, 423 HCC samples were selected for further study. Next, we matched gene expression profiles of PROM1 (CD133), RALYL and TGFB2 with corresponding survival data. To ensure the accuracy, samples with an OS or PFS < 1 month were excluded from survival analysis. After a careful review, a total of 393 samples were included. Patients were then classified into low- and high-expression groups based on the optimal cut-off value determined by X-Tile software. Kaplan–Meier (K-M) survival curves were used to analyze the differences in survival time between low- and high-expression patients. K-M survival curves were plotted by using the ‘survival’ package in the R environment (Version 3.5.2). OS time of clinical samples from TCGA database and their corresponding expression of CD133, TGF-β2 and RALYL were shown in the Supporting Excel form 1.

GEPIA2 is a user-friendly platform to perform survival analyses based on gene expression. The thresholds for high/low expression level cohorts can be adjusted by users (Tang et al., 2017; Tang et al., 2019). However, we do not know how to find the optimal cut-off value using GEPIA. We notice that the sample size of the TCGA patient cohort consistently changed with different cut-offs, as showed by the arrows in the below figures (Figure 3 and 4). Sometimes, it is shown that the sample size is insufficient at adjusted thresholds so that the analysis cannot be completed (Figure 5 below). The relative low expression level of stemness-related genes in tumor tissue may be responsible for the failure.

We hope our reply can clarify this issue.

Figure 3

GEPIA 2

Survival Analysis

In this pane, you can perform survival analysis based on the expression status of one gene or a multi-gene signature and plot a Kaplan-Meier curve. Some gene signature lists are provided. Example

Survival Analysis | Most Differential Survival Genes | Survival Map

Survival Plots

GEPIA performs overall survival (OS) or disease free survival (DFS, also called relapse-free survival and RFS) analysis based on gene expression. GEPIA uses Log-rank test, a.k.a the Mantel-Cox test, for hypothesis test. Cohorts thresholds can be adjusted, and gene-pairs can be used. The Cox proportional hazard ratio and the 95% confidence interval information can also be included in the survival plot.

Parameters

- Gene: Input a gene/isoform of interest.
- Normalized by gene: Set the gene/isoform used for normalizing in "Gene" field.
- Methods: Select the OS or DFS survival method.
- Axis Units: Select Month or Day unit for plotting.
- Datasets Selection/Datasets: Select one or multiple cancer types of interest in the "Dataset Selection" field and click "add" to build dataset list in the "Datasets" field. Also, manual input of cancer types split by comma (e.g. COAD,READ) is also acceptable. Meanwhile, users can also choose cancer subtype (For example, the Breast Lumina A subtype) by clicking the "Subtype Filter" check-box.
- Color Reverse: Choose whether to reverse the default color.
- Group Cutoff: Select a suitable expression threshold for splitting the high-expression and low-expression cohorts.
- Cutoff-High(%): Samples with expression level higher than this threshold are considered as the high-expression cohort.
- Cutoff-Low(%): Samples with expression level lower than this threshold are considered as the low-expression cohort.

--- Help ---

Gene Signatures
Gene A
CD133
Input a gene symbol or id.
Normalized by gene (optional)
Methods
Overall Survival
Disease Free Survival (RFS)
Group Cutoff
Median
Quartile
Custom
Cutoff-High(%)
50
Cutoff-Low(%)
50

Hazards Ratio (HR)
Yes
Calculate the hazards ratio based on Cox PH Model.
95% Confidence Interval
Yes
Add the 95% CI as dotted line.
Axis Units
Months
High Group
Low Group

Multiple Datasets Subtype Filter
Datasets Selection (Cancer name)
Tips: Ctrl/Command + A: select all cancer types.
GBM
HNSC
KICH
KIRC
KIRP
LAML
LGG
LIHC
Datasets
Add | Reset
The plot axis x order will follow the list.
Plot

Figure 4

Figure 5

Comments 5:

Of note, it would be desirable that the sequencing data is deposited in a publically accessible database and made available not only on “reasonable request”.

Our reply:

Yes, we already upload the sequencing data on the Gene Expression Omnibus (GEO) under accession GSE163601.

Comments 6:

It is unclear why the authors selected a different gating strategy for CD133. This might dilute the fraction of putative CSCs and, thus, not be meaningful.

Our reply:

In original manuscript, we acquired higher percentage of CD133⁺ cells in MIHA than that in our published work (Ma et al., Gastroenterology, 2007.). The reviewer wanted us to explain the difference in percentage of CD133⁺ cells observed in this study. We believed that the gating strategy for CD133 accounted for the difference. Thus, we adjusted gate setting in accord with our published paper (Ma et al., Gastroenterology, 2007.) and the percentage of CD133⁺ cells was similar in our two works (this study and Ma et al., Gastroenterology, 2007.).

Actually, the aim of this study was to compare the CD133⁺ proportion in parental HCC cells with RALYL-transfected ones. The gate setting is entirely same when it was applied for flow cytometry of parental HCC cells and RALYL-transfected ones. So, the conclusion could not be affected by gating strategy. The adjusted gating strategy is used to be in accord with our previous work (Ma et al., Gastroenterology, 2007.). Thanks for your questions and hope our reply can answer your question.

Comments 7:

The lack of validation of the findings in freshly isolated hepatocytes remains a significant shortcoming that could be overcome, e.g. using organoid cultures.

Our reply:

Thanks for the Reviewer's advices. However, it was really difficult for us to isolate fresh human hepatocytes for organoid culture. Alternatively, organoid derived from mouse fresh hepatocytes were applied in our study. To investigate the role of RALYL in normal hepatocytes, full-length mouse RALYL cDNA (mRALYL) was cloned into a lentiviral vector and then stably transfected into the mouse normal liver organoid. After transfection, more and larger organoids were observed in mRALYL group than control group (Supplementary Fig. 4c). In addition, mRALYL overexpression in normal mouse hepatocytes could upregulate stemness-related genes (Supplementary Fig. 1c).

Reviewer #3 (Remarks to the Author):

This work demonstrates that RALYL is a critical regulator of HCC cancer stem cell populations via its effects on TGF-B2 mRNA stability and PI3K/AKT and STAT3 signaling. The authors also present intriguing data suggesting that this effect on mRNA stability could be mediated via N6-methyladenosine in the TGF-B2 3'UTR. My primary concerns with the initial submission were that I could not interpret the western blots through which the authors aimed to demonstrate changes in markers of EMT upon overexpression or knockdown of RALYL, and that there was insufficient evidence to support the model that the observed change in RALYL stability is regulated by N6-methyladenosine (m6A). With respect to the first point, the authors have repeated the western blots so that they can now be interpreted much more easily. They have also added a few functional experiments testing the possibility that m6A may be involved, including assays that suggest interactions between RALYL and TGF-B2, and between RALYL and the m6a demethylase, FTO. In addition, they have addressed the minor points that I brought up, and I think the manuscript is appropriate for publication in its current revised form.

Our reply:

Thanks!

Reviewers' Comments:

Reviewer #1:

Remarks to the Author:

The authors experimentally addressed all major concerns and clarified several issues. The revision significantly strengthened the study. I have no further comment.

A point-by-point response to the reviewers' comments

First review:

Reviewer #1

The study addresses an important aspect of cancer stemness in primary liver cancer, and it is technically well performed. The manuscript is well structured and written. However, potential translational implications should be validated in independent patient cohorts. Furthermore, mechanistic analyses should be extended, particularly related to the role of RALYL as bona fide oncogene and the association to TGF- β 2.

Major comments:

1. Adjacent liver tissue has very similar expression signature like HCC tissue. To conclusively judge the reliability of the model isolated hepatocytes and non-diseased livers should be included in Figure 1. Adjacent tumor tissue seems not suitable as a normal control given the altered microenvironment and mix of different cell types. Furthermore, selection of the markers should be explained in more detail. Is this a pre-selected list of genes (reference) or were these genes identified by unsupervised methods? Statistical pipeline should be delineated (Figure 1B).

Our reply:

Thanks to the reviewer's comments. We agree with the reviewer's suggestion that non-diseased liver is more appropriate, but it is difficult for us to collect sample and perform the transcriptome sequencing during the current situation of Covid-19 pandemic. We will use healthy liver in our further study.

According to literature, the representative specific markers are selected for each liver development stage (Aaron M. Zorn, liver development, StemBook, 2008). The pre-selected list of genes was used to investigate whether the *in vitro* liver development model used in our study is reliable. The RNA-seq results indicated that most of the markers are indeed highly expressed in their corresponding stage, therefore, we believed the *in vitro* liver development model is reliable. As the reviewer raised this issue, it seems a bit confusing to show too many markers in Figure 1b. Therefore, the number of markers has been narrowed down to 3-4 genes for each stage (see revised Figure 1b).

2. Expression level of CSC and stemness related markers (CD133, CD24, C-MYC) should have also been demonstrated at different stages of hepatocyte differentiation,

not just for established hepatoma and immortalized hepatocytes cell lines (Supplementary Figure 1B and C). Result should support the finding that high expression of RALYL in LP and HL is followed by increased expression of these markers, particularly CD133.

Authors should also more clearly delineate how and why RALYL gene was selected as a potential candidate gene with implications for cancer stemness. This is unclear as presented.

Our reply:

According to Reviewer's suggestion, expression pattern of HCC stemness related markers CD133, C-MYC and EpCAM were tested by qRT-PCR in our *in vitro* liver development model. The results showed that CD133 and EpCAM were highly expressed in LP and PH as RALYL (see revised Supplementary Figure 1a). Furthermore, the expressions of CD133, C-MYC and EpCAM were also increased in RALYL-transfected PLC-8024 and MIHA cells (see revised Supplementary Figure 1c). This information has been added to the revised "Results" section (the first paragraph, page #5).

Sorry for the unclear statement in our original manuscript about how and why RALYL gene was selected. In the present study, we aim to identify and characterize genes related to the stemness maintenance and regulation. The selection priority includes gene(s) encoding nuclear protein which show similar expression pattern to liver progenitor markers. Several genes including RALYL were selected and transfected into immortalized liver cell line MIHA and HCC cell line PLC-8024. Spheroid formation assay was used to assess the self-renew ability of those candidate genes, and RALYL showed strongest spheroid formation ability. In addition, we found that RALYL showed strongest ability to upregulate stemness markers, such as CD133, AFP, C-MYC and NANOG. Therefore, RALYL was selected for further study. This information has been added to the revised "Results" section (the first paragraph, page #5).

3. Univariate analyses of 117 HCC samples showed that patients with higher RALYL expression positively correlated with poorer differentiation state, vascular invasion, metastases, as well as poor overall survival. This finding should further be supported with the data regarding CD133 expression in these tumors as well as TGF- β 2 status.

Furthermore, in concordance with REMARK guidelines, multivariate analyses should be performed to confirm independent prognostic implications. In line with this, results should be confirmed in an independent cohort of authentic HCC, e.g. publically available datasets.

Our reply:

In accordance with reviewer's suggestion, we also assessed the expression of CD133 and TGF- β 2 in the 117 HCC cases. Association study found that high CD133 expression was significantly associated with poor differentiation stage ($P=0.034$) and metastasis ($P=0.050$), and high expression of TGF- β 2 was positively correlated with poor differentiation stage ($P=0.009$), vascular invasion ($P=0.014$) and metastasis ($P=0.024$) (Table 2 and Table 3). As the expression of RALYL was positively correlated with poorer differentiation state, vascular invasion and metastases, there were quite consistent among these genes in clinical significance in our cohort. In addition, the high expression of RALYL was significantly associated with higher expression of CD133 ($P=0.003$) and TGF- β 2 ($P=0.029$) (revised Table 1). Cox proportional hazard regression analysis showed that RALYL expression was an independent prognostic factor for overall survival of HCC patients ($P=0.046$; Table 4). In terms of survival, the expression CD133 was closely associated with poor prognosis in our in-house cohort ($P=0.021$) as well as TCGA database ($P=0.004$). Similarly, higher expression of TGF- β 2 is associated with poorer overall survival rate in both our in-house cohort ($P=0.002$) as well as TCGA database ($P=0.047$) (revised Supplementary Figure 1f). In line with the prognostic implications of RALYL in 117 patient samples, high expression of RALYL is closely associated with poor overall survival rate ($P=0.006$) in TCGA database (revised Figure 1f). All these results indicate that RALYL, CD133 and TGF- β 2 have similar clinical implications. This information has been added into the revised 'Results' and 'Discussion' section (The second paragraph, Page #6; the first paragraph, Page #14).

4. Tumorigenicity of two "normal" (L02 and MiHA) hepatocyte cell lines after forced expression of RALYL is potentially interesting since this would establish RALYL as a bone fide oncogene. However, this requires more thorough investigations and should be extended (e.g. induction in isolated hepatocytes). In addition, previous studies showed that both MiHA and LO2 do not express CD133 (PMID 17570225). This should be explained. In addition, both Huh7 (mutation) and Hep3B (null) show alterations in

p53. What is the status in LO2 and MIHA? It would be interesting to include e.g. HepG2 as p53 WT cell line to exclude p53-dependent mechanisms.

Our reply:

We agree with reviewer's points that the function roles of RALYL as a bone fide oncogene should be extended in isolated hepatocytes. We have endeavored to isolate fresh hepatocytes in adjacent non-tumor liver tissues and transfect RALYL into the hepatocytes. Unfortunately, it is really difficult to do the transfection and passage the fresh hepatocytes to perform in *in vitro* functional studies.

Yes, according to our previous work (Ma et al., Gastroenterology, 2007.), CD133 is hardly detected in MIHA and LO2. Because of the gate setting in flow cytometry, we acquired higher percentage of CD133⁺ cells in the original manuscript. By resetting the gate in accord with our previous paper, the percentage of CD133⁺ cells in MIHA was similar in our two works (see revised Figure 1d).

MIHA and LO2 possess wild-type TP53(Yamamoto et al., Cancers, 2019. Li et al., BMC cancer, 2017.), therefore, molecular biology experiments performed in those two cell lines could exclude the p53-dependent mechanism.

5. Difference in proliferation upon transfection with RALYL is not striking. Significant difference is present only at some time points (Figure 2B and Supplementary Figure 2B).

Our reply:

Thanks for the reviewers' comments. Two more independent experiments for each cell line were performed with three repeats. In revised manuscript, all data of three independent experiments with error bar has been presented. Although it is not striking at some time points, difference between experiment and control group is significant at statistical level ($P < 0.05$). The P values were shown in revised Figure 2B and Supplementary Figure 2B.

6. Western blot results, particularly for overexpression, are not of sufficient quality and unconvincing as presented (SLUG, Fibronectin, E-Cadherin). Quality should be improved (Figure 3D). Also applies to Figure 5B – some western blots are of insufficient quality and difficult to interpret. This should be improved.

Our reply:

Thanks for the comments. We have repeated western blotting. Western blotting images with high quality replaced the original ones in revised Figure 3D and Figure 5B.

7. The authors should explain the selection of the utilized chemotherapeutic agents. It would be important to explore approved first line therapies for HCC (i.e. sorafenib, lenvatinib). This should be included.

Our reply:

Hepatic arterial infusion chemotherapy (HAIC) with 5-fluorouracil (5-FU) and cisplatin (CDDP) is commonly used for advanced hepatocellular carcinoma (HCC). (Park et al., Cancer, 2007. Ueshima et al., Oncology, 2010.) Thus 5-Fu and CDDP were used as chemotherapeutic agents in our present study. Currently, sorafenib and lenvatinib are the approved first line therapy for HCC. In the present study, we found that RALYL could increase drug resistance for 5-Fu and CDDP treatment, but not for sorafenib or lenvatinib (see revised supplement Figure 4c for sorafenib results, lenvatinib results were not shown).

8. Mechanistic investigations, in particular the role of TGF- β 2 are potentially interesting and relevant. However, majority of results were obtained by Hep3B, Huh7, MIHA and LO2. It is unclear why 8024 was used for the transcriptome analyses. At least one more cell line should be included in the analyses. It would also be important demonstrate effect on target genes after silencing of RALYL expression.

Our reply:

MIHA, LO2 and PLC-8024 were used for RALYL overexpressing system, while Huh7, Hep3B and H2M were used for RALYL knockdown experiments. PLC-8024 was used for all functional and molecular experiments in our study (Figure 1, 2, 3, 4, 5 and 6, Supplementary Figure 1, 4 and 5). At first, PLC-8024 was used for the transcriptome analyses to unveil the mechanism underlying RALYL over-expression. It was found that TGF- β 2 as well as several genes was up-regulated in RALYL over-expressing cells. Thereafter, qRT-PCR was performed to detect the expression level of TGF- β 2 and other genes in Hep3B, Huh7, MIHA and LO2. The up-regulation of TGF- β 2 can be observed in PLC-8024, MIHA, LO2 with RALYL overexpression, consistently, the expression level of TGF- β 2 was decreased in RALYL knockdown Huh7, Hep3B and H2M cells (See in revised Supplementary Figure 5C). As qRT-PCR results are consistent in all HCC cells, transcriptome analysis of more HCC cells

was not carried out. Due to the Covid-19 pandemic, it's also hard for us to perform transcriptome sequencing for other cell lines.

Minor comments

1. Supplementary Figure 1 – correct figure legend (Four instead of three cell lines).

Our reply:

Thanks! We have corrected it accordingly. (See in Supplementary Figure 1)

2. Axis on flow cytometry dot plots should be adequately labeled (Supplementary Figure 4E)

Our reply:

The Axis on flow cytometry in Supplementary Figure 4E has been adequately labeled (See in Supplementary Figure 4e).

3. Intrasplenic tumor injections are frequently limited by tumor growth within the spleen (Andersen et al. Sci Transl. Med 2010). Please comment.PMID: 20962331

Our reply:

We understand the Reviewer's concern. Intrasplenic injection is a widely used as liver metastasis model (Endo et al., Cancer Gene Therapy, 2002.). Some studies indicated that the metastasis in liver of nude mouse did not affected by the tumor formation in the spleens (Giavazzi et al., Cancer Research, 1986.). In our study, visible tumor formation in spleen was not commonly observed and only detected in 4/27 of mice intrasplenic injected with different cells. Therefore, we believe that being a HCC metastatic mouse model, intrasplenic injection is better than tail vein injection which often causes pulmonary metastasis (Kozlowski et al, Cancer Research, 1984).

Reviewer #2 (Remarks to the Author):

Overall I think this work is solid and of interest, and I do not see major experiments that would need to be included for publication. I do, however, think that the conclusion that m6A is involved is premature (explained below), so I would suggest either removing those data or performing additional experiments to validate this conclusion. Some points that should be addressed prior to publication are listed below.

Significant points:

1. It is a little bit unclear whether the differentiation model described here is new to this report or not. In the main text it is referred to as new, but in the methods a paper in press is referenced. This may just be a consequence of delays and multiple papers being under review at the same time, but this should be clarified in the revised manuscript.

Our reply:

The liver differentiation model has been studied by RNA-seq in another paper (Ming Liu et al., PNAS, 2020). In that paper, we described how to establish the *in vitro* liver differentiation model and how to discover two subtypes of liver cancer with different oncofetal properties and therapeutic targets using that model. When we submitted this manuscript to *Nature Communications*, that work was under revision. In this work, RALYL was selected as a stemness-related gene for further study.

2. The poor quality of many of the western blots in figure 3D make it difficult to assess the authors claims about EMT markers. The results in 3A-C support the claim that RALYL induces migration and invasion, so perhaps this could be removed (and the text adjusted) if alternative expression data is not available.

Our reply:

Sorry for the poor quality of western blotting images. According to Reviewer's suggestion, we have repeated the western blotting. Images were replaced with high quality ones in revised Figure 3D.

3. Please clarify what is meant by the statement that "a rare expression change was detected in the control cells (Figure 4D and Supplementary Figure 4C)." (it would be helpful to specifically indicate which lanes we should be comparing).

Our reply:

Sorry for the confusing statement in the original manuscript. It is indeed difficult to distinguish the difference between atRA treatment and no treatment groups by conventional PCR results. Therefore, qRT-PCR was performed to compare the fold change of the mature hepatocyte markers and the stemness markers between RALYL overexpression and control groups. We found that the fold changes of expressions of mature hepatocyte markers (CK8, CK18, and albumin) were significantly higher in 8024-RALYL and MIHA-RALYL cell after atRA treatment, compared with 8024-Vec and MIHA-Vec. Whereas, the fold change of expressions of stemness-associated genes

(AFP, CD133, and NANOG) and RALYL were significantly lower in 8024-RALYL and MIHA-RALYL cells compared with controls (Figure 4d), suggesting that RALYL-expressing cells possessed higher differentiation potential than controls. The qRT-PCR results were shown in revised Figure 4D.

4. While there does, indeed, seem to be a difference in stability in TGFB mRNA when RALYL is overexpressed, the model that this stability is regulated through m⁶A would require more experimental validation than the m⁶A-IP that is presented (particularly given the fact that m⁶A antibodies have been shown to recognize related modifications). Since this is a relatively minor point in the paper, I would suggest removing the m⁶A data. If the authors wish to pursue this further, they would need additional experiments to validate the involvement of m⁶A, such as testing the effects of m⁶A-regulatory proteins (METTL3/METTL14, YTHDF2 etc), and/or more detailed mapping of putative m⁶A sites. The authors should also consider other possible mechanisms of mRNA stability regulation.

Our reply:

Thanks to the reviewer for the comment. According to Reviewer's suggestion, additional experiments were performed to validate the m⁶A modification by RALYL. As is known, m⁶A modification usually presents in the consensus sequence RRACH and the enrichment of m⁶A in 3' untranslated regions (3' UTRs) near stop codon is reported to regulate the stability of mRNA (Zhao et al., Molecular Cell Biology, 2017). We found that multiple RRACH sites were located in 3' UTR of TGF-β2 mRNA around stop codon. The luciferase reporter assays confirmed the interaction between RALYL and 3' -UTR of TGF-β2 mRNA as the relative luciferase activity in RALYL-overexpressing cells was significantly augmented compared with control cells. Furthermore, we investigated whether RALYL could cooperative two well-known m⁶A erasers, including Fat mass and obesity-associated protein (FTO) and alkB homologue 5 (ALKBH5) and the m⁶A reader YTHDF2. IP assays are performed to confirm the interaction between RALYL and several well-known m⁶A erasers in exogenously expressed stable 8024-RALYL-flag and MIHA-RALYL-flag cells. It was excited to find that FTO is pulled by flag-RALYL, but ALKBH5 and YTHDF2 were not detected in immunoprecipitation. Thus, we concluded that RALYL could cooperate with FTO

to remove m⁶A of TGF-β2 mRNA and keep its stability. All those results were added to revised manuscript (The first paragraph, Page #16).

Minor points:

1. Labeling is not always totally clear. For instance, the heat map in 1B has the HL abbreviation, but this doesn't seem to be defined anywhere.

Our reply:

Sorry for the language errors. We have revised the abbreviations in Figure 1b. The full names for induced human embryonic stem (ES) cells, definitive endoderm (DE), liver progenitor cells (LP) and premature hepatocytes (PH) were added in revised figure legend.

2. The 'RNA stability' methods section may need to be reworded a bit – as written it appears that the Actinomycin D treatment was for 48 hours when I believe the authors meant to state that the time course of samples (0 hr -> 10 hrs) was collected after cells had been transfected for 48 hours.

Our reply:

Thank you for the reviewer's comments. We have rewritten the 'RNA stability' methods section. The revised section was seen in the first paragraph, Page #20.

3. Some of the figures are presented in a confusing format. For instance, in supplementary figure 1A-C, qPCR data are presented as connected lines as if to indicate a continuous variable on the x axis. While the argument could be made that the progression from ES HEP does represent a time-like variable, this is definitely not the case in panel 1C where the x-axis simply lists different target genes.

Our reply:

Thanks for reviewer's kind reminder. Yes, we totally agree with the reviewer's suggestions. We have replaced the line chart by bar chart to show qPCR data (see in revised supplementary figure 1A-C).

Second review

Reviewer #3 (Remarks to the Author) (reviewer #3 commented on our response to the original reviewer #1 (unable to re-review)):

This is the revised version of the manuscript by Wang et al.

The authors experimentally addressed several of the raised concerns and clarified several of the comments. The revision strengthened the study. However, several other major concerns have only been partly or not addressed by the authors. In particular, translational implications are only completely delineated.

Comments 1:

The study still misses non-diseased liver tissue as controls for the study. The provided answer related to the COVID19 situation does not suffice. While the impact of the pandemic for translational science is enormous and certainly appreciated, archived tissue can be utilized or the authors could establish collaborations to address this critical issue. In addition, preliminary investigations using publically available data would be a viable option.

Our reply:

According to Reviewer's suggestion, RNA-seq was performed with two normal liver specimens. In general, the expression profiles are quite similar between normal liver (NL) and hepatocytes (HEP) adjacent to HCC tissues, especially for those selected genes in our original manuscript (Fig. 1b), which is shown in this letter as Figure 1. qRT-PCR was also applied to validate the expression level of some well-known markers of each stage and results showed similar expression level of these markers in HEP and NL (original Supplementary Fig. 1a, Figure 2 in this letter). In the revised Fig. 1b and Supplementary Fig 1a, selected genes in HEPs were replaced by NLs (see revised Fig. 1b, revised Supplementary Fig. 1a).

Figure 1

Figure 2

Comments 2:

While it is still unclear why the selection of genes was reduced to only few genes despite availability of whole transcriptome data, the reviewer appreciates that the graphical representation in Figure 1 is now quite intuitive.

Our reply:

The purpose of this study is to identify and characterize genes related to the stemness maintenance and regulation in HCC cancer stem cells (CSCs) from our recently established *in vitro* hepatocyte differentiation model. As described in the manuscript, this model includes embryonic stem (ES) cells, definitive endoderm (DE), liver progenitor (LP) cells and premature hepatocytes (PH). RNA-seq was performed with these four stage cells, as well as two pairs of HCC clinical specimens (HEP and HCC). A total of 20,038 protein-coding genes were detected by RNA-seq. 2,324 genes were selected for those highly expressed in LP cells compared to ES, EN, PH cells and clinical samples. Next, 126 genes which expression level are much higher in HCC cells (> 5 fold) compared with non-tumor cells were selected. 34 genes encoding nuclear proteins were then selected. Based on literature review, 9 not-well studied genes were selected for further study. Spheroid formation assay was used to study the self-renew ability of these candidate genes in immortalized liver cell line MIHA and HCC cell line PLC-8024, and RALYL showed strongest spheroid formation ability. In addition, RALYL also showed the strongest ability to up-regulate stemness markers, such as CD133, AFP, C-MYC and NANOG. Therefore, RALYL was further investigated in the present study.

Comments 3:

The prognostic implications are still not quite clear. How were the variables for the multivariate analyses selected? Why are essential clinico-pathological associations from Table 1 as well as CD133 and TGFB2 not included?

Our reply:

According to Reviewer's suggestion, we included more related variables such as 'cirrhosis', 'vascular invasion', 'CD133' and 'TGFB2' expression to the multivariate analyses. Results showed that RALYL is an independent prognostic indicator for the survival of HCC patients. This result has been added in the revised manuscript (revised Table 2).

Comments 4:

Furthermore, it is unclear how the TCGA data was retrieved and analyzed. Validation of the findings by GEPIA2 could not confirm the presented findings for CD133, RALYL and TGFB2. Please explain and provide the pipeline of the analyses.

Our reply:

The process of the TCGA data retrieval and analysis are as follows:

1) Data retrieval: The data of the TCGA dataset is freely accessible. Efforts by centers (e.g. the University of California Santa Cruz) allow users to easily access the normalized TCGA RNA-seq, methylation and clinical data in excel files. We retrieved and downloaded normalized RNA-seq data and corresponding clinical information of the Liver Cancer (LIHC) cohort from the TCGA hub in UCSC XENA website

[https://xenabrowser.net/datapages/?cohort=TCGA%20Liver%20Cancer%20\(LIHC\)&removeHub=https%3A%2F%2Fxcena.treehouse.gi.ucsc.edu%3A443](https://xenabrowser.net/datapages/?cohort=TCGA%20Liver%20Cancer%20(LIHC)&removeHub=https%3A%2F%2Fxcena.treehouse.gi.ucsc.edu%3A443)).

2) Data analysis: After screening, 423 HCC samples were selected for further study. Next, we matched gene expression profiles of PROM1 (CD133), RALYL and TGFB2 with corresponding survival data. To ensure the accuracy, samples with an OS or PFS < 1 month were excluded from survival analysis. After a careful review, a total of 393 samples were included. Patients were then classified into low- and high-expression groups based on the optimal cut-off value determined by X-Tile software. Kaplan–Meier (K-M) survival curves were used to analyze the differences in survival time between low- and high-expression patients. K-M survival curves were plotted by using the 'survival' package in the R environment (Version 3.5.2). OS time of clinical samples from TCGA database and their corresponding expression of CD133, TGF-β2 and RALYL were shown in the Supporting Excel form 1.

GEPIA2 is a user-friendly platform to perform survival analyses based on gene expression. The thresholds for high/low expression level cohorts can be adjusted by users (Tang et al., 2017; Tang et al., 2019). However, we do not know how to find the optimal cut-off value using GEPIA. We notice that the sample size of the TCGA patient cohort consistently changed with different cut-offs, as showed by the arrows in the below figures (Figure 3 and 4). Sometimes, it is shown that the sample size is

insufficient at adjusted thresholds so that the analysis cannot be completed (Figure 5 below). The relative low expression level of stemness-related genes in tumor tissue may be responsible for the failure.

We hope our reply can clarify this issue.

Figure 3

Figure 4

Figure 5

Comments 5:

Of note, it would be desirable that the sequencing data is deposited in a publically accessible database and made available not only on “reasonable request”.

Our reply:

Yes, we already upload the sequencing data on the Gene Expression Omnibus (GEO) under accession GSE163601.

Comments 6:

It is unclear why the authors selected a different gating strategy for CD133. This might dilute the fraction of putative CSCs and, thus, not be meaningful.

Our reply:

In original manuscript, we acquired higher percentage of CD133⁺ cells in MIHA than that in our published work (Ma et al., Gastroenterology, 2007.). The reviewer wanted us to explain the difference in percentage of CD133⁺ cells observed in this study. We believed that the gating strategy for CD133 accounted for the difference. Thus, we adjusted gate setting in accord with our published paper (Ma et al., Gastroenterology, 2007.) and the percentage of CD133⁺ cells was similar in our two works (this study and Ma et al., Gastroenterology, 2007.).

Actually, the aim of this study was to compare the CD133⁺ proportion in parental HCC cells with RALYL-transfected ones. The gate setting is entirely same when it was applied for flow cytometry of parental HCC cells and RALYL-transfected ones. So, the conclusion could not be affected by gating strategy. The adjusted gating

strategy is used to be in accord with our previous work (Ma et al., Gastroenterology, 2007.). Thanks for your questions and hope our reply can answer your question.

Comments 7:

The lack of validation of the findings in freshly isolated hepatocytes remains a significant shortcoming that could be overcome, e.g. using organoid cultures.

Our reply:

Thanks for the Reviewer's advices. However, it was really difficult for us to isolate fresh human hepatocytes for organoid culture. Alternatively, organoid derived from mouse fresh hepatocytes were applied in our study. To investigate the role of RALYL in normal hepatocytes, full-length mouse RALYL cDNA (mRALYL) was cloned into a lentiviral vector and then stably transfected into the mouse normal liver organoid. After transfection, more and larger organoids were observed in mRALYL group than control group (Supplementary Fig. 4c). In addition, mRALYL overexpression in normal mouse hepatocytes could upregulate stemness-related genes (Supplementary Fig. 1c).

Reviewer #2 (Remarks to the Author):

This work demonstrates that RALYL is a critical regulator of HCC cancer stem cell populations via its effects on TGF-B2 mRNA stability and PI3K/AKT and STAT3 signaling. The authors also present intriguing data suggesting that this effect on mRNA stability could be mediated via N6-methyladenosine in the TGF-B2 3'UTR. My primary concerns with the initial submission were that I could not interpret the western blots through which the authors aimed to demonstrate changes in markers of EMT upon overexpression or knockdown of RALYL, and that there was insufficient evidence to support the model that the observed change in RALYL stability is regulated by N6-methyladenosine (m6A). With respect to the first point, the authors have repeated the western blots so that they can now be interpreted much more easily. They have also added a few functional experiments testing the possibility that m6A may be involved, including assays that suggest interactions between RALYL and TGF-B2, and between RALYL and the m6a demethylase, FTO. In addition, they have addressed the minor points that I brought up, and I think the manuscript is appropriate for publication in its current revised form.

Our reply:

Thanks!